

**Impacts of microtopographic snow-redistribution and lateral subsurface processes**
**on hydrologic and thermal states in an Arctic polygonal ground ecosystem**
**Gautam Bisht[1], William J. Riley[1], Haruko M. Wainwright[1], Baptiste Dafflon[1], Yuan**
**Fengming[2], and Vladimir E. Romanovsky[3]**
[1]Climate & Ecosystem Sciences Division, Lawrence Berkeley National Laboratory,1
Cyclotron Road, Berkeley, California 94720, USA
[2]Environmental Sciences Division, Oak Ridge National Laboratory, Oak Ridge, TN, 37831-
6301 , USA
[3]Geophysical Institute, University of Alaska Fairbanks, Fairbanks, AK 99775, USA
Correspondence to: Gautam Bisht (gbisht@lbl.gov)
**Abstract**

18       Microtopographic features, such as polygonal ground, are characteristic sources of

landscape heterogeneity in the Alaskan Arctic coastal plain. Here, we analyze the effects of
snow redistribution (SR) and lateral subsurface processes on hydrologic and thermal states
at a polygonal tundra site near Barrow, Alaska. We extended the land model integrated in
the ACME Earth System Model (ESM) to redistribute incoming snow by accounting for
microtopography and incorporated subsurface lateral transport of water and energy
(ALMv0-3D). Three 10-years long simulations were performed for a transect across
polygonal tundra landscape at the Barrow Environmental Observatory in Alaska to isolate
the impact of SR and subsurface process representation. When SR was included, model
results show a better agreement (higher $R^2$ with lower bias and RMSE) for the observed
differences in snow depth between polygonal rims and centers. The model was also able to
accurately reproduce observed soil temperature vertical profiles in the polygon rims and
centers (overall bias, RMSE, and $R^2$ of $0.59^0C$, $1.82^0C$, and 0.99, respectively). The spatial
heterogeneity of snow depth during the winter due to SR generated surface soil



temperature heterogeneity that propagated in depth and time and led to ~10 cm shallower
and ~5 cm deeper maximum annual thaw depths under the polygon rims and centers,
respectively. Additionally, SR led to spatial heterogeneity in surface energy fluxes and soil
moisture during the summer. Excluding lateral subsurface hydrologic and thermal
processes led to small effects on mean states but an overestimation of spatial variability in
soil moisture and soil temperature as subsurface liquid pressure and thermal gradients
were artificially prevented from spatially dissipating over time. The effect of lateral
subsurface processes on active layer depths was modest with mean absolute difference of
~3 cm. Our integration of three-dimensional subsurface hydrologic and thermal subsurface
dynamics in the ACME land model will facilitate a wide range of analyses heretofore
impossible in an ESM context.
**1    Introduction**

The northern circumpolar permafrost region, which contains ~1700 Pg of organic

carbon down to 3 m (Tarnocai et al., 2009), is predicted to experience disproportionately
larger future warming compared to the tropics and temperate latitudes (Holland and Bitz,
2003). Recent warming in the Arctic has led to changes in lake area (Smith et al., 2005),
snow cover duration and extent (Callaghan et al., 2011a), vegetation cover (Sturm et al.,
2005), growing season length (Smith et al., 2004), thaw depth (Schuur et al., 2008),
permafrost stability (Jorgenson et al., 2006), and land-atmosphere feedbacks (Euskirchen
et al., 2009). Future predictions of Arctic warming include northward expansion of shrub
cover in tundra (strum 2001, Tape et al 2006), decreases in snow cover duration
(Callaghan et al., 2011a), and emissions of $CO_2$ and $CH_4$ from decomposition of
belowground soil organic matter (Koven et al., 2011; Schaefer et al., 2011; Schuur and
Abbott, 2011,Xu, 2016 #154; Xu et al., 2016).

Several recent modeling studies have predicted a positive carbon-climate feedback

at the global scale (Cox et al., 2000; Dufresne et al., 2002; Friedlingstein et al., 2001; Fung et
al., 2005; Govindasamy et al., 2011; Jiang et al., 2011; Jones et al., 2003; Koven et al., 2015;
Matthews et al., 2007b; Matthews et al., 2005; Sitch et al., 2008; Thompson et al., 2004;
Zeng et al., 2004), although the strength of this predicted feedback at the year 2100 was





shown to have a large variability across models (Friedlingstein et al., 2006). In contrast to
the ocean carbon cycle, the terrestrial carbon cycle is expected to be a more dominant
factor in the global carbon-climate feedback over the next century (Matthews et al., 2007a;
Randerson et al., 2015).

Changes in Arctic ecosystem net ecosystem productivity (NEP, defined as the

difference between net primary production (NPP) and heterotrophic respiration ($R_h$)) will
be determined by the magnitude and direction of changes in NPP and $R_h$. Warming
experiments in the Arctic have found increases and decreases of plant growth in response
to higher temperatures (Barber et al., 2000; Chapin et al., 1995; Cornelissen et al., 2001;
Hobbie and Chapin, 1998; Hollister et al., 2005; Van Wijk et al., 2004; Walker et al., 2006;
Wilmking et al., 2004). Arctic ecosystems are limited in nitrogen availability (Schimel et al.,
1996; Shaver and Chapin III, 1986) and higher mineralization rates under warmer climate
(Hobbie, 1996) could lead to higher $CO_2$ fixation by plants (Shaver and Chapin, 1991).
Additionally, a longer growing season is expected to result in a negative carbon-climate
feedback by increasing NPP (Euskirchen et al., 2006). On the other hand, microbial
decomposition of previously frozen soil organic matter under a warmer climate is expected
to strengthen the carbon-climate feedback (Davidson and Janssens, 2006; Mack et al., 2004;
Oechel et al., 1993; Tarnocai et al., 2009).

Snow, which covers the Arctic ecosystem for 8-10 months each year (Callaghan et

al., 2011b), is a critical factor influencing hydrologic and ecologic interactions (Jones,
1999). Snowpack modifies surface energy balances (via high reflectivity), soil thermal
regimes (due to low thermal conductivity), and hydrologic cycles (because of melt water).
Several studies have shown that warm soil temperatures under snowpack support the
emission of greenhouse gases from belowground respiration (Grogan and Chapin Iii, 1999;
Sullivan, 2010) and nitrogen mineralization (Borner et al., 2008; Schimel et al., 2004)
during winter. Additionally, decreases in snow cover duration have been shown to increase
net ecosystem $CO_2$ uptake (Galen and Stanton, 1995; Groendahl et al., 2007). Recent snow
manipulation experiments in the Arctic have provided evidence of the importance of snow
in the expected responses of Arctic ecosystems under future climate change (Morgner et al.,
2010; Nobrega and Grogan, 2007; Rogers et al., 2011; Schimel et al., 2004; Wahren et al.,
2005; Welker et al., 2000).





Apart from the spatial extent and duration of snowpack, the spatial heterogeneity of
snow depth is an important factor in various terrestrial processes (Clark et al., 2011;
Lundquist and Dettinger, 2005). The spatial distribution of snow not only affects the
quantity of snowmelt discharge (Hartman et al., 1999; Luce et al., 1998), but also the water
chemistry (Rohrbough et al., 2003; Wadham et al., 2006; Williams et al., 2001). Lawrence
and Swenson (2011) demonstrated the importance of snow depth heterogeneity in
predicting responses of the Arctic ecosystem to future climate change by performing
idealized numerical simulations of shrub expansion across the pan-Arctic region using the
Community Land Model (CLM4). Their results showed that an increase in active layer
thickness (ALT) under shrubs was negated when spatial heterogeneity in snow cover due
to wind driven snow redistribution was accounted for, resulting in an unchanged grid cell
mean active layer thickness. López-Moreno et al. (2014) identified processes responsible
for snow depth heterogeneity at three distinct spatial scales: microtopography at 1-10 m
(Lopez-Moreno et al., 2011); wind induced lateral transport processes at 100-1000 m
(Liston et al., 2007); and precipitation variability at catchment scales of 10 – 1000 km
(Sexstone and Fassnacht, 2014).
Large portions of the Arctic are characterized by polygonal ground features, which
are formed in permafrost soil when frozen ground cracks due to thermal contraction
during winter and ice wedges form within the upper several meters (Hinkel et al., 2005).
Polygons can be classified as 'low-centered' or 'high-centered' based on the relationship
between their central and mean elevations. Polygonal ground features are dynamic
components of the Arctic landscape in which the upper part of ice-wedge thaw under low-
centered polygon troughs leads to subsidence, eventually ($\sim$o(centuries)) converting the
low-centered polygon into a high-centered polygon (Seppala et al., 1991). Microtopography
of polygonal ground influences soil hydrologic and thermal conditions (Engstrom et al.,
2005). In addition to controlling $CO_2$ and $CH_4$ emissions, soil moisture affects (1)
partitioning of incoming radiation into latent, sensible, and ground heat fluxes (Hinzman
and Kane, 1992; McFadden et al., 1998); (2) photosynthesis rates (McGuire et al., 2000;
Oberbauer et al., 1991; Oechel et al., 1993; Zona et al., 2011); and (3) vegetation
distributions (Wiggins, 1951).



Our goals in this study include (1) analyzing the effects of spatially heterogeneous
snow in polygonal ground on soil temperature and moisture and surface processes (e.g.,
surface energy budgets); (2) analyzing how model predictions are affected by inclusion of
lateral subsurface hydrologic and thermal processes; and (3) developing and testing a
three-dimensional version of the land model ALM (Tang and Riley, 2016; Zhu and Riley,
2015) integrated in the ACME Earth System Model (ESM). We note that the original version
of ALM is equivalent to CLM4.5 (Koven et al., 2013; Oleson, 2013a), and represents vertical
energy and water dynamics, including phase change. We expanded on that model to
explicitly represent soil lateral energy and hydrological exchanges and fine-resolution
snow redistribution (ALMv0-3D). We then applied ALMv0-3D to a transect across a
polygonal tundra landscape at the Barrow Environmental Observatory in Alaska. After
defining our study site, the model improvements, model tests against observations, and
analyses, we apply the model to examine the effects of snow redistribution and lateral
subsurface processes on snow micro-topographical heterogeneity, soil temperature, and
the surface energy budget.

## 2   Methodology

### 2.1   Study Area

Our analysis focuses on sites located near Barrow, Alaska (71.3⁰ N, 156.5⁰ W) from
the long term Department of Energy (DOE) Next-Generation Ecosystem Experiment (NGEE-
Arctic) project. The four primary NGEE-Arctic study sites (A, B, C, D) are located within the
Barrow Environmental Observatory (BEO), which is situated on the Alaskan Coastal Plain.
The annual mean air temperature for our study sites is approximately -13°C (Walker et al.,
2005) and mean annual precipitation is 106 mm with the majority of precipitation
occurring during the summer season (Wu et al., 2013). The study site is underlain with
continuous permafrost (Brown et al., 1980) and the annual maximum thaw depth (active
layer depth) ranges between 30-90 cm (Hinkel et al., 2003). Although the overall
topographic relief for the BEO is low, the four NGEE study sites have distinct
microtopographic features: low-centered (A), high-centered (B), and transitional polygons
(C, D). Contrasting polygon types are indicative of different stages of permafrost





degradation and were the primary motivation behind the choice of study sites for the
NGEE-Arctic project. LIDAR Digital Elevation Model (DEM) data were available at 0.25 m
resolution for the region encompassing all four NGEE sites. In this work, we perform
simulations along a two-dimensional transect in low-centered polygon Site-A as shown by
the dotted line in Figure 1.

## 2.2  ALMv0 Description

We developed the capability to represent three-dimensional hydrology and thermal
dynamics in ALMv0 (Zhu et al., 2016b), and call the new model ALMv0-3D. ALMv0 was
derived from CLM4.5 (Ghimire et al., 2016; Koven et al., 2013), and is the land model
integrated in the ACME Earth System Model (ESM). The model represents coupled plant
biophysics, soil hydrology, and soil biogeochemistry (Oleson *et al.* 2013). We run ALMv0-
3D here with prescribed plant phenology (called Satellite Phenology (SP) mode), since our
focus is on the thermal dynamics of the system, rather than the C cycle dynamics.

## 2.3  Representing Two- and Three-Dimensional Physics

### 2.3.1  Subsurface hydrology

The flow water in the unsaturated zone is given by the $\theta$-based Richards equations
as

$$\frac{\partial \theta}{\partial t} = -\nabla \cdot \vec{q} - Q \tag{1}$$

where $\theta$ [m³m⁻³] is the volumetric soil water content, $t$ [s] is time, $\vec{q}$ [ms⁻¹] is Darcy flux, and
$Q$ [m of water m⁻³ of soil s⁻¹] is volumetric sink of water. Darcy flux is given by

$$\vec{q} = -k\nabla(\psi + z) \tag{2}$$

where $k$ [ms⁻¹] is the hydraulic conductivity and $\psi$ [m] is the soil matric potential. The
hydraulic conductivity and soil matric potential are non-linear functions of volumetric soil
moisture. ALMv0 uses the modified form of Richards equation of Zeng and Decker (2009)
that computes Darcy flux as

$$\vec{q} = -k\nabla(\psi + z - C) \tag{3}$$

where C is a constant hydraulic potential above the water table, $z_\nabla$, given as



$$C = \psi_E + z = \psi_{sat}\left[\frac{\theta_E(z)}{\theta_{sat}}\right]^{-B} + z = \psi_{sat} + z_\nabla \tag{4}$$

where $\psi_E$ [m] is the equilibrium soil matric potential. Substituting equations (3) and (4)
into equation (1) yields the equation for the vertical transport of water in ALMv0:

$$\frac{\partial \theta}{\partial t} = \frac{\partial}{\partial z}\left[k\left(\frac{\partial(\psi - \psi_E)}{\partial z}\right)\right] - Q \tag{5}$$

A finite volume spatial discretization and implicit temporal discretization with Taylor
series expansion leads to a tri-diagonal system of equations. We extended this 1-D Richards
equation to a 3-D representation integrated in ALMv0-3D, which is presented next.

We use a cell-centered finite volume discretization to decompose the spatial domain

into $N$ non-overlapping control volumes, $\Omega_n$, such that $\Omega = \cup_{n=1}^{N}\Omega_n$ and $\Gamma_n$ represents the
boundary of the $n$-th control volume. Applying a finite volume integral to equation (1) and
the divergence theorem yields

$$\frac{\partial}{\partial t}\int_{\Omega_n}\theta dV = -\int_{\Gamma_n}\left(\vec{q}\cdot d\vec{A}\right) - \int_{\Omega_n}Q dV \tag{6}$$

The spatially discretized equation for the $n$-th grid cell that has $V_n$ volume and $n'$ neighbors
is given by

$$\frac{d\theta_n}{dt}V_n = -\sum_{n'}\left(\vec{q}_{nn'}\cdot\vec{A}_{nn'}\right) - QV_n \tag{7}$$

For the sake of simplicity in presenting the discretized equation, we assume the 3-D grid is
a Cartesian grid with each grid cell having a thickness of $\Delta x$, $\Delta y$, and $\Delta z$ in the $x$, $y$, and $z$
directions, respectively. Using an implicit time integral, the 3-D discretized equation at time
$t+1$ for a $(i,j,k)$ control volume is given as

$$\left(\frac{\Delta\theta_{i,j,k}^{t+1}}{\Delta t}\right)V_{i,j,k} = \left(q_{x_{i-1/2,j,k}}^{t+1} - q_{x_{i+1/2,j,k}}^{t+1}\right)\Delta y\Delta z$$

$$+ \left(q_{y_{i,j-1/2,k}}^{t+1} - q_{y_{i,j+1/2,k}}^{t+1}\right)\Delta x\Delta z$$

$$+ \left(q_{z_{i,j,k-1/2}}^{t+1} - q_{z_{i,j,k+1/2}}^{t+1}\right)\Delta x\Delta y - QV_{i,j,k} \tag{8}$$

where $q_x$, $q_y$ and $q_z$ are Darcy flux in the $x$, $y$, and $z$ directions, respectively and $\Delta\theta_{i,j,k}^{t+1}$ is the
change in volumetric soil liquid water in time $\Delta t$. Using the same approach as Oleson





(2013b), the Darcy flux in all three directions is linearized about $\theta$ using Taylor series
expansion. The linearized Darcy flux in the $x$ direction at the $(i - 1/2, j, k)$ interface is a
function of $\theta_{i-1,j,k}$ and $\theta_{i,j,k}$:

$$q_{x_{i-1/2,j,k}}^{t+1} = q_{x_{i-1/2,j,k}}^{t} + \frac{\partial q_{x_{i-1/2,j,k}}^{t}}{\partial \theta_{i-1,j,k}} \Delta\theta_{i-1,j,k}^{t+1} + \frac{\partial q_{x_{i-1/2,j,k}}^{t}}{\partial \theta_{i,j,k}} \Delta\theta_{i+1,j,k}^{t+1} \qquad (9)$$

The linearized Darcy fluxes in the $y$ and $z$ directions are computed similarly. Substituting
equation (9) in equation (8) results in a banded matrix of the form

$$\alpha\Delta\theta_{i-1,j,k}^{t+1} + \beta\Delta\theta_{i,j-1,k}^{t+1} + \gamma\Delta\theta_{i,j,k-1}^{t+1} + \eta\Delta\theta_{i+1,j,k}^{t+1} + \mu\Delta\theta_{i,j+1,k}^{t+1} + \phi\Delta\theta_{i,j,k+1}^{t+1}$$
$$+ \zeta\Delta\theta_{i,j,k}^{t+1} = \varphi \qquad (10)$$

where $\alpha$, $\beta$, and $\gamma$ are subdiagonal entries; $\eta$, $\mu$, and $\phi$ are superdiagonal entries; $\zeta$ is
diagonal entry of the banded matrix; and $\varphi$ is a column vector given by

$$\alpha = \frac{\partial q_{x_{i-1/2,j,k}}^{t}}{\partial \theta_{i-1,j,k}} \Delta y \Delta z \qquad (11)$$

$$\beta = \frac{\partial q_{y_{i,j-1/2,k}}^{t}}{\partial \theta_{i,j-1,k}} \Delta x \Delta z \qquad (12)$$

$$\gamma = \frac{\partial q_{z_{i,j,k-1/2}}^{t}}{\partial \theta_{i,j,k-1}} \Delta x \Delta y \qquad (13)$$

$$\eta = \frac{\partial q_{x_{i+1/2,j,k}}^{t}}{\partial \theta_{i+1,j,k}} \Delta y \Delta z \qquad (14)$$

$$\mu = \frac{\partial q_{y_{i,j+1/2,k}}^{t}}{\partial \theta_{i,j+1,k}} \Delta x \Delta z \qquad (15)$$

$$\phi = \frac{\partial q_{z_{i,j,k+1/2}}^{t}}{\partial \theta_{i,j,k+1}} \Delta x \Delta y \qquad (16)$$

$$\zeta = \left(\frac{\partial q_{x_{i-1/2,j,k}}^{t}}{\partial \theta_{i,j,k}} - \frac{\partial q_{x_{i+1/2,j,k}}^{t}}{\partial \theta_{i,j,k}}\right)\Delta y\Delta z + \left(\frac{\partial q_{y_{i,j-1/2,k}}^{t}}{\partial \theta_{i,j,k}} - \frac{\partial q_{y_{i,j+1/2,k}}^{t}}{\partial \theta_{i,j,k}}\right)\Delta x\Delta z$$
$$+ \left(\frac{\partial q_{z_{i,j-1/2,k}}^{t}}{\partial \theta_{i,j,k}} - \frac{\partial q_{z_{i,j+1/2,k}}^{t}}{\partial \theta_{i,j,k}}\right)\Delta x\Delta y - \frac{\Delta x\Delta x\Delta z}{\Delta t} \qquad (17)$$





$$\varphi = -\left(q_{x_{i-\frac{1}{2},j,k}}^{t} - q_{x_{i+\frac{1}{2},j,k}}^{t}\right)\Delta y \Delta z - \left(q_{y_{i,j-\frac{1}{2},k}}^{t} - q_{y_{i,j+\frac{1}{2},k}}^{t}\right)\Delta x \Delta z$$

$$-\left(q_{z_{i,j-\frac{1}{2},k}}^{t} - q_{z_{i,j+\frac{1}{2},k}}^{t}\right)\Delta x \Delta y + Q_{i,j,k}^{t+1}\Delta x \Delta x \Delta z \tag{18}$$

The coefficients of equation (10) described in equation (11)-(18) are for an internal grid
cell with six neighbors. The coefficients for the top and bottom grid cells are modified for
infiltration and interaction with the unconfined aquifer in the same manner as Oleson
(2013b). Similarly, the coefficients for the grid cells on the lateral boundary are modified
for a no-flux boundary condition. See Oleson (2013b) for details about the computation of
hydraulic properties and derivative of Darcy flux with respect to soil liquid water content.

### 2.3.2  Subsurface thermal

ALMv0 solves a tightly coupled system of equations for soil, snow, and standing
water temperature (Oleson, 2013a). The model solves the transient conservation of energy:

$$c\frac{\partial T}{\partial t} = -\nabla \cdot F \tag{19}$$

where $c$ is the volumetric heat capacity [J m$^{-3}$ K$^{-1}$], F is the heat flux [W m$^{-2}$], and t is time
[s]. The heat conduction flux is given by

$$F = -\lambda \nabla T \tag{20}$$

where $\lambda$ is thermal conductivity [W m$^{-1}$ K$^{-1}$] and T is temperature [K]. Applying a finite
volume integral to equation (20) and divergence theorem yields

$$c\frac{\partial}{\partial t}\int_{\Omega_n} T = -\int_{\Gamma_n} \vec{F} \cdot d\vec{A} \tag{21}$$

The spatially discretized equation for a $n$-th grid cell that has $V_n$ volume and $n'$ neighbors is
given by

$$c_n\frac{dT_n}{dt}V_n = -\sum_{n'}\left(\vec{F}_{nn'} \cdot \vec{A}_{nn'}\right) \tag{22}$$

Similar to the approach taken in Section 2.3.1, ALMv0-3D assumes a 3-D Cartesian grid
with each grid cell having a thickness of $\Delta x$, $\Delta y$, and $\Delta z$ in the $x$, $y$, and $z$ directions,
respectively. Temporal integration of equation (22) is carried out using the Crank-
Nicholson method that uses a linear combination of fluxes evaluated at time t and t + 1:


$$c_n \frac{\left(T_{i,j,k}^{t+1} - T_{i,j,k}^{t}\right)}{\Delta t} \Delta x \Delta y \Delta z$$

$$= \omega \left\{ \left(F_{x_{i-1/2,j,k}}^{t} - F_{x_{i+1/2,j,k}}^{t}\right) \Delta y \Delta z \right.$$

$$+ \left(F_{y_{i,j-1/2,k}}^{t} - F_{y_{i,j+1/2,k}}^{t}\right) \Delta x \Delta z$$

$$+ \left(F_{z_{i,j,k-1/2}}^{t} - F_{z_{i,j,k+1/2}}^{t}\right) \Delta x \Delta y \right\}$$

$$+ (1-\omega) \left\{ \left(F_{x_{i-1/2,j,k}}^{t+1} - F_{x_{i+1/2,j,k}}^{t+1}\right) \Delta y \Delta z \right.$$

$$+ \left(F_{y_{i,j-1/2,k}}^{t+1} - F_{y_{i,j+1/2,k}}^{t+1}\right) \Delta x \Delta z$$

$$+ \left(F_{z_{i,j,k-1/2}}^{t+1} - F_{z_{i,j,k+1/2}}^{t} + 1\right) \Delta x \Delta y \right\} \tag{23}$$

where $\omega$ is the weight in the Crank-Nicholson method and set to 0.5 in this study.
Substituting a discretized form of heat flux using equation (20) in equation (23), results in
a banded matrix of the form

$$\alpha T_{i-1,j,k}^{t+1} + \beta T_{i,j-1,k}^{t+1} + \gamma T_{i,j,k-1}^{t+1} + \eta T_{i+1,j,k}^{t+1} + \mu T_{i,j+1,k}^{t+1} + + \phi T_{i,j,k+1}^{t+1} + \zeta \Delta T_{i,j,k}^{t+1}$$

$$= \varphi \tag{24}$$

where $\alpha$, $\beta$, and $\gamma$ are subdiagonal entries; $\eta$, $\mu$, and $\phi$ are superdiagonal entries; $\zeta$ is
diagonal entry of the banded matrix; and $\varphi$ is a column vector given by

$$\alpha = \left(\frac{-\omega' \Delta t}{c_{i,j,k} \Delta x}\right) \left(\frac{\lambda_{i-1/2,j,k}}{x_{i,j,k} - x_{i-1,j,k}}\right) \tag{25}$$


$$\beta = \left(\frac{-\omega' \Delta t}{c_{i,j,k} \Delta y}\right) \left(\frac{\lambda_{i,j-1/2,k}}{y_{i,j,k} - y_{i-1,j,k}}\right) \tag{26}$$


$$\gamma = \left(\frac{-\omega' \Delta t}{c_{i,j,k} \Delta z}\right) \left(\frac{\lambda_{i,j,k-1/2}}{z_{i,j,k} - z_{i,j,k-1}}\right) \tag{27}$$


$$\mu = \left(\frac{-\omega' \Delta t}{c_{i,j,k} \Delta x}\right) \left(\frac{\lambda_{i+1/2,j,k}}{x_{i+1,j,k} - x_{i,j,k}}\right) \tag{28}$$


$$\xi = \left(\frac{-\omega' \Delta t}{c_{i,j,k} \Delta y}\right) \left(\frac{\lambda_{i-1/2,j,k}}{y_{i+1,j,k} - y_{i,j,k}}\right) \tag{29}$$




$$\phi = \left(\frac{-\omega'\Delta t}{c_{i,j,k}\Delta z}\right)\left(\frac{\lambda_{i-1/2,j,k}}{z_{i+1,j,k} - z_{i,j,k}}\right)$$ (30)


$$\zeta = 1 + \left(\frac{\omega'\Delta t}{c_{i,j,k}\Delta x}\right)\left[\frac{\lambda_{i-1/2,j,k}}{x_{i,j,k} - x_{i-1,j,k}} + \frac{\lambda_{i+1/2,j,k}}{x_{i+1,j,k} - x_{i,j,k}}\right]$$
$$+ \left(\frac{\omega'\Delta t}{c_{i,j,k}\Delta y}\right)\left[\frac{\lambda_{i,j-1/2,k}}{y_{i,j,k} - y_{i-1,j,k}} + \frac{\lambda_{i-1/2,j,k}}{y_{i+1,j,k} - y_{i,j,k}}\right]$$
$$+ \left(\frac{\omega'\Delta t}{c_{i,j,k}\Delta z}\right)\left[\frac{\lambda_{i,j,k-1/2}}{z_{i,j,k} - z_{i,j,k-1}} + \frac{\lambda_{i-1/2,j,k}}{z_{i+1,j,k} - z_{i,j,k}}\right]$$ (31)


$$\varphi = T_{i,j,k}^t + \left(\frac{\omega\Delta t}{c_{i,j,k}\Delta x}\right)\left(F_{x_{i-1/2,j,k}}^t - F_{x_{i+1/2,j,k}}^t\right)$$
$$+ \left(\frac{\omega\Delta t}{c_{i,j,k}\Delta y}\right)\left(F_{y_{i,j-1/2,k}}^t - F_{y_{i,j+1/2,k}}^t\right)$$
$$+ \left(\frac{\omega\Delta t}{c_{i,j,k}\Delta z}\right)\left(F_{z_{i,j,k-1/2}}^t - F_{z_{i,j,k+1/2}}^t\right)$$ (32)

The coefficients of equation (24) described in equation (25)-(32) are for an internal grid
cell with six neighbors. The coefficients for the top and bottom grid cells are modified for
presence of snow and/or standing water, and no-flux boundary. The coefficients for the
grid cells on the lateral boundary are modified for a no-flux boundary condition. ALM
handles ice-liquid phase transitions by first predicting temperatures at the end of a time
step and then updating temperatures after accounting for deficits or excesses of energy
during melting or freezing. See Oleson (2013b) for details about the computation of
thermal properties and phase transition.
**2.3.3  Numerical solution via PETSc**
ALMv0, which considers flow only in the vertical direction, solves a tridiagonal and
banded tridiagonal system of equations for water and energy transport, respectively. In
ALMv0-3D, accounting for lateral flow in the subsurface results in a sparse linear system,
equations (10) and (24), where the sparcity pattern of the linear system depends on grid
cell connectivity. In this work, we use the PETSc (Portable, Extensible Toolkit for Scientific



Computing) library (Balay et al., 2016) developed at the Argonne National Laboratory to
solve the sparse linear systems. PETSc provides object-oriented data structures and solvers
for scalable scientific computation on parallel supercomputers.

### 2.4   Snow Model and Redistribution

The snow model in ALMv0-3D is the same as that in the default ALMv0 and CLM4.5
(Anderson, 1976; Dai and Zeng, 1997; Jordan, 1991). The snow model allows for a dynamic
snow depth and up to 5 snow layers, and explicitly solves the vertically-resolved mass and
energy budgets. Snow aging, compaction, and phase change are all represented in the snow
model formulation. Additionally, the snow model accounts for the influence of aerosols
(including black and organic carbon and mineral dust) on snow radiative transfer (Oleson,
2013a). ALMv0 uses the methodology of Swenson and Lawrence (2012) to compute
fractional snow cover area, which is appropriate for ESM-scale grid cells (~100 [km] x 100
[km]). Since the grid cell resolution in this work is sub-meter, we modified the fractional
cover to be either 1 (when snow was present) or 0 (when snow was absent). Two main
drivers of snow redistribution (SR) include topography and surface wind (Warscher et al.,
2013); previous SR models include mechanistically- (Bartelt and Lehning, 2002; Liston and
Elder, 2006) and empirically- (Frey and Holzmann, 2015; Helfricht et al., 2012) based
approaches. To mimic the effects of wind, we used a conceptual model to simulated SR over
the fine-resolution topography of our site by instantaneously re-distributing the incoming
snow flux such that lower elevation areas (polygon center) receive snow before higher
elevation areas (polygon rims). This relatively simple and parsimonious approach is
reasonable given the observed snow depth heterogeneity, as described below, and small
spatial extent of our domain.

### 2.5   System Characterization

Hydrologic and thermal properties differ by depth and landscape type. We used the
horizontal distribution of OM organic matter from Wainwright et al. (2015) to infer soil
hydrologic and thermal properties following the default representations in ALM.
Vegetation cover was classified as arctic shrubs in polygon centers and arctic grasses in
polygon rims. The default representation of the plant wilting factor assigns a value of zero





for a given soil layer when it's temperature falls below a threshold ($T_{threshold}$) of -2 $^0$C. This
default value leads to overly large predicted latent and sensible heat fluxes during winter,
compared to nearby eddy covariance measurements. We modified $T_{threshold}$ to be 0 $^0$C in this
study, resulting in improved predicted wintertime latent heat fluxes compared to the
default version of the model (**Error! Reference source not found.**). Although biases
compared to the observations remain, particularly for sensible heat fluxes in the spring, the
improvement is substantial and, given the observational uncertainties, we believe sufficient
to justify our use of the model for investigations of the role of snow heterogeneity in this
polygonal tundra system.
## 2.6   Simulation Setup, Climate Forcing, and Analyses

Because of computational constraints, we investigated the role of snow

redistribution and physics representation using a two-dimensional transect through site A
(Figure 1). The transect was 104 [m] long and 45 [m] deep that was discretized
horizontally with a grid spacing of 0.25 [m] and an exponentially varying layer thickness in
the vertical with 30 soil layers. No flow conditions for mass and energy were imposed on
the east, west, and bottom boundaries of the domain. Temporal discretization of 30 [min]
was used in the simulations. All simulations were performed in "SP" mode, i.e., Leaf Area
Index (LAI) was prescribed from MODIS observations.

Simulations were run for 10 years using long-term climate data gathered at the

Barrow, Alaska Observatory site (https://www.esrl.noaa.gov/gmd/obop/brw/) managed
by the Global Monitoring Division of NOAA's Earth System Research Laboratory (Mefford et
al., 1996). The missing precipitation time series was gap-filled using daily precipitation at
the Barrow Regional Airport available from the Global Historical Climatology Network
(http://www1.ncdc.noaa.gov/pub/data/ghcn/daily). We tested the model by comparing
predictions to high-frequency observations of snow depth and vertically resolved soil
temperature for September 2012 – September 2013.  Temperature observations were
taken at discrete locations in a polygon center and rim (Figure 1), and were combined to
analyze comparable landscape positions in the simulations (Figure 2).

After testing, the model was used to investigate the effect of snow redistribution and

2D subsurface hydrologic and thermal physics by analyzing three scenarios: (1) no snow



redistribution and 1D physics; (2) snow redistribution and 1D physics; and (3) snow
redistribution and 2D physics. Between these scenarios, we compared vertically-resolved
soil temperature and liquid saturation, active layer depth, and mean and spatial variation of
latent and sensible heat fluxes across the 10 years of simulations. For each soil column, the
simulated soil temperature was interpolated vertically and the active layer depth was
estimated as the maximum depth that had above-freezing soil temperature.

## 3    Results and Discussion

### 3.1    Snow depth

In the absence of SR, predicted snow depth exactly follows the topography. With SR,

a much larger dependence of winter-average snow depth on topography is predicted
(Figure 2). Further, for the winter average, there are very small differences in snow depth
between simulations with SR and 1D or 2D subsurface physics representations. Compared
to observations, considering snow redistribution led to: (1) a factor of ~2 improvement in
snow depth bias for the polygon center; (2) modest increase and decrease in average bias
on the rims for September through February and March through June, respectively; and (3)
a dramatic improvement in bias of the difference in snow depth between the polygon
centers and rims (Figure 3). There was no discernible difference in snow depth bias
between the 1D and 2D physics (Table 1), although the predicted subsurface temperature
fields were different, as shown below.

The temporal variation of the mean snow depth (**Figure 4**a) and its spatial standard

deviation (**Figure 4**b) also differed based on whether SR was considered, but was not
affected by considering 2D thermal or hydrologic physics. With SR, the snow depth
coefficient of variation (**Figure 4**c) was about 0.5 from December through the beginning of
the snowmelt period, indicating relatively large spatial heterogeneity. Snapshots of
simulated snow depth for the three simulation scenarios are included in Supplementary
material (**Error! Reference source not found.**).



### 3.2 Soil Temperature and Active Layer Depth


Broadly, ALMv0-3D accurately predicted the polygon center soil temperature at
depth intervals corresponding to the temperature probes (0-20 cm, 20-50 cm, 50-75 cm,
and 75-100 cm; Figure 5a). Recall that the observed temperatures for the polygon center
and rims were taken at single points in site A (Figure 1) while the predicted temperatures
were calculated as averages across the transect for each of the two landscape position
types. The model was able to simulate early freeze up of the soil column under the rims as
compared to centers in November 2012 because of differences in accumulated snow pack.
The transition to thawed soil in the 0-20 cm depth interval in early June 2013 and the
subsequent temperature dynamics over the summer were very well captured by ALMv0-
3D. Minimum temperatures during the winter were also accurately predicted, although the
temperatures in the deepest layer (75-100 cm) were overestimated by ~3°C in March. For
figure clarity we did not indicate the standard deviation of the observations, but provide
that information in Supplemental Material (**Error! Reference source not found.** - **Error!**
**Reference source not found.**).
Similarly, the soil temperatures were accurately predicted in the polygon rims
(Figure 5b). The largest discrepancies between measured and predicted soil temperatures
were in the shallowest layer (0 - 25 cm), where the predictions were up to a few °C cooler
than some of the observations between December 2012 and March 2013. In the polygon
center, a thicker snow pack acts as a heat insulator and keeps soil temperature higher in
winter as compared to the polygon rims.
Three recent studies have used other mechanistic models to simulate the soil
temperature fields at this site, and achieved comparably good comparisons with the
observations (Kumar et al. 2016 applied a 3D version of PFLOTRAN; Atchley et al. 2015 and
Harp et al. 2016 applied a 1D version of ATS). However, those models used the measured
soil temperature near the surface as the top boundary condition. In contrast, the top
boundary condition in this work is the climate forcing (air temperature, wind, solar
radiation, humidity, precipitation), and the ground heat flux is prognosed based on ALM's
vegetation and surface energy dynamics. We note that no parameter calibration was done




in this work or that of Kumar et al. (2016), while the ATS parameterizations were tuned to
match the soil temperature profile.
Snow redistribution impacts spatial variability of soil temperature throughout the
soil column. Absence of SR results in no significant spatial variability of soil temperature
(Figure 6a). Inclusion of SR on the surface modifies the amount of energy exchanged
between the snow and the top soil layer, thereby creating spatial variability in the
temperature of the top soil, which propagates down into the soil column (Figure 6b). With
SR, energy dissipation in the lateral direction reduces the penetration depth of the soil
temperature spatial variance (compare Figure 6c and Figure 6b).
With 1D physics, the average spatial and temporal difference of the active layer
depth (ALD) between simulations with and without SR was 1.7 cm (Figure 7a), and the
absolute difference was 6.5 cm. As described above, we diagnosed the ALD to be the
maximum soil depth during the summer at which vertically interpolated soil temperature
is 0 °C. On average, the rims had ~10 cm shallower ALD with (blue line) than without
(green line) SR, consistent with the loss of insulation from SR on the rims during the
winter. In the centers (e.g., at location 42 - 55 m), the thaw depth was deeper by ~5 cm
with SR because of the higher snow depth there from SR. The effect of SR on the ALD was
largest on the rims because, compared to centers, they (1) on average lost more snow with
SR and (2) are more thermally conductive. Since rims are therefore colder at the time of
snowmelt with SR, the ground heat flux during the subsequent summer was unable to thaw
the soil column as deeply as when SR is ignored. For comparison, Atchley et al. (2015)
found in their sensitivity analysis using the 1D version of ATS that SR resulted in deeper
thaw depths in both polygon centers (by ~3 cm) and rims (~0.3 cm). Thus, there results for
polygon centers are consistent in sign but lower in magnitude than ours, but opposite in
sign for the rims.
Across ten years of simulation, the inter-annual variability (IAV) in ALD varied
substantially between the three scenarios (Figure 7b). As expected, for the 1D physics
without SR scenario (green line), the IAV in ALD was determined by landscape position
because of differences in soil and vegetation parameters. With SR and 1D physics, the
model shows largest differences over the rims, again highlighting the relatively larger
effects of SR on the rim soil temperatures.



The effect of 1D versus 2D physics on the ALD across the transect was modest
(mean absolute difference ~3 cm). Generally, because 2D physics allows for lateral energy
diffusion, the horizontal variation of ALD was slightly lower (i.e., the red line is smoother
than the blue line; Figure 7a) than with 1D physics. This difference was also reflected in the
thaw depth IAV across the transect, where 2D physics led to a smoother lateral profile of
inter-annual variability than with 1D physics.
The impact of physics formulation (i.e., 1D or 2D) alone was investigated by
analyzing differences between soil temperature profiles over time for polygon rims and
centers in simulations with snow redistribution. Inclusion of 2D subsurface physics
resulted in soil temperatures with depth and time that were lower in the polygon rims
(Figure 8a) and higher in polygon centers (Figure 8b). Using the simulations from the
scenario with SR and 2D physics, we evaluated the extent to which the soils under rims and
centers can be separately considered as relatively homogeneous single column systems by
evaluating the soil temperature standard deviation as a function of depth and time (Figure
9). During winter, both polygon rims and centers showed soil temperature spatial
variability >1 °C up to a depth of ~2 [m]. The soil temperature spatial variability in winter
due to snow redistribution is dissipated over the summer. During the summer, polygon
centers were relatively more homogeneous vertically compared to polygon rims.

### 3.3   Surface Energy Budget

Predicted monthly- and spatial-mean ($\mu$) surface latent heat fluxes across the
transect were very similar between the three scenarios (Figure 10a), with a growing
season mean difference of < 1.0 [W m$^{-2}$]. However, the spatial variability (SV = $\sigma$; Figure
10b) and coefficient of variation (CV = $\sigma/\mu$; Figure 10c) of latent heat fluxes were different
between the scenarios with SR (1D and 2D physics) and without SR. With SR, the latent
heat flux spatial standard deviation peaked after snowmelt and declined until the fall when
snow began, from about ~100% to 10% of the mean. This relatively larger spatial variation
in latent heat flux occurred because of large spatial heterogeneity in near surface soil
moisture in the beginning of summer, indicating a residual effect of SR from the previous
winter.



The predicted temporal monthly-mean and spatial-mean surface sensible heat

fluxes across the transect were also similar between the three scenarios (Figure 11a), with
a growing season mean absolute difference of < 3.5 W m$^{-2}$. Also, the sensible heat flux
spatial variability differences occurred earlier than snowmelt, in contrast to the latent heat
flux. Both the standard deviation and CV of the sensible heat fluxes were larger than those
of the latent heat fluxes, with early season standard deviations of ∼50 W m$^{-2}$ (Figure 11b)
and CV's of ∼1.5 (Figure 11c). As for the latent heat fluxes, the differences in standard
deviation and CV of sensible heat fluxes were small between the 1D and 2D scenarios with
SR, arguing that the subsurface lateral energy exchanges associated with the 2D physics did
not propagate to the mean surface heat fluxes. However, as for the latent heat flux, there
was a relatively large difference in spatial variation between the scenarios with and
without SR (e.g., of about 25 W m$^{-2}$ in May; Figure 10b).
**3.4   Soil Moisture**

Neither SR nor 2D lateral physics affected the spatial mean moisture across time

(not shown). However, the spatial heterogeneity of predicted soil moisture content differed
substantially between scenarios during the snow free period (Figure 12). For the 1D
simulations, the effect of SR was to increase the growing season soil moisture spatial
heterogeneity by factors of 5.2 and 1.6 for 0-10 cm and 10-65 cm depth intervals,
respectively  (compare Figure 12a and Figure 12b). Compared to the 1D physics, simulating
2D thermal and hydrologic physics led to an overall reduction in the soil moisture spatial
heterogeneity by factors of 0.8 and 0.7 for 0-10 cm and 10-65 cm depth intervals,
respectively (compare Figure 12b and Figure 12c). Thus, with respect to dynamic spatial
mean soil moisture, SR effects dominated those associated with lateral subsurface water
movement .
**3.5   Caveats and Future Work**

The good agreement between ALMv0-3D predictions and soil temperature

observations demonstrate the model's capabilities to represent this very spatially
heterogeneous and complex system. However, several caveats to our conclusions remain
due to uncertainties in model parameterizations, model structure, and climate forcing data.



Because of computational constraints, we applied a 2D transect domain to the site,
instead of a full 3D domain. We are working to improve the computational efficiency of the
model, which will facilitate a thorough analysis of the effects of 3D subsurface energy and
water fluxes. A related issue is our simplified treatment of surface water flows. A thorough
analysis of the effects of surface water redistribution would require integration of a 2D
surface thermal flow model with the ALMv0-3D in a 3D domain, which is another goal for
our future work. However, we note that the good agreement using the 2D model domain
supports the idea that a two-dimensional simplification may be appropriate for this system.
The expected geomorphological changes in these systems over the coming decades (e.g.,
Liljedahl et al. 2016), which will certainly affect soil temperature and moisture, are not
currently represented in ALM, although incorporation of these processes is a long-term
development goal.
The current representation of vegetation in ALMv0-3D for these polygonal tundra
systems is over-simplified. For example, non-vascular plants (mosses and lichens) are not
explicitly represented in the model, but can be responsible for a majority of evaporative
losses (Miller et al., 1976) and are strongly influenced by near surface hydrologic
conditions (Williams and Flanagan, 1996). Our use of the 'satellite phenology' mode, which
imposes transient LAI profiles for each plant functional type in the domain, ignores the
likely influence of nutrient constraints (Zhu et al., 2016a) on photosynthesis and therefore
the surface energy budget. Other model simplifications, e.g., the simplified treatment of
radiation competition may also be important, especially as simulations are extended over
periods where vegetation change may occur (e.g., Grant 2016).

## 4    Summary and Conclusions

We analyzed the effects of microtopographical surface heterogeneity and lateral
subsurface transport in a polygonal tundra landscape on soil temperature, soil moisture,
and surface energy exchanges. Starting from the climate-scale land model ALMv0, we
incorporated in ALMv0-3D numerical representations of subsurface water and energy
lateral transport that are solved using PETSc. A simple method for redistributing incoming
snow along the microtopographic transect was also integrated in the model.

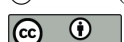


Over the observational record, ALMv0-3D with snow redistribution and lateral heat
and hydrological fluxes accurately predicted snow depth and soil temperature vertical
profiles in the polygon rims and centers (overall bias, RMSE, and $R^2$ of $0.59^0$C, $1.82^0$C and
0.99, respectively). In the rims, the transition to thawed soil in spring, summer
temperature dynamics, and minimum temperatures during the winter were all accurately
predicted. In the centers, a ~2℃ warm bias in April in the 75-100 cm soil layer was
predicted, although this bias disappeared during snowmelt.
The spatial heterogeneity of snow depth during the winter due to snow
redistribution generated surface soil temperature heterogeneity that propagated into the
soil over time. The temporal and spatial variation of snow depth was affected by snow
redistribution, but not by lateral thermal and hydrologic transport. Both snow
redistribution and lateral thermal fluxes affected spatial variability of soil temperatures.
Energy dissipation in the lateral direction reduced the depth to which soil temperature
variance penetrated. Snow redistribution led to ~10 cm shallower active layer depths
under the polygon rims because of the residual effect of reduced insulation during the
winter. In contrast, snow redistribution led to ~5 cm deeper active layers under the
polygon centers. The effect of lateral energy fluxes on active layer depths was ~3 cm.
Compared to 1D physics, the 2D subsurface physics led to lower (higher) soil temperatures
with depth and time in the polygon rims (centers). The larger than 1 ℃ wintertime spatial
temperature variability down to ~2 m depth in rims and centers indicates the uncertainty
associated with considering rims and centers as separate 1D columns. During the summer,
polygon center temperatures were relatively more vertically homogeneous than
temperatures in the rims.
The monthly- and spatial-mean predicted latent and sensible heat fluxes were
unaffected by snow redistribution and lateral heat and hydrological fluxes. However, snow
redistribution led to spatial heterogeneity in surface energy fluxes and soil moisture during
the summer. Excluding lateral subsurface hydrologic and thermal processes led to an over
prediction of spatial variability in soil moisture and soil temperature because subsurface
gradients were artificially prevented from laterally dissipating over time.    Snow
redistribution effects on soil moisture heterogeneity were larger than those associated
with lateral thermal fluxes.
Overall, our analysis demonstrates the potential and value of explicitly representing
snow redistribution and lateral subsurface hydrologic and thermal dynamics in polygonal
ground systems and quantifies the effects of these processes on the resulting system states
and surface energy exchanges with the atmosphere. The integration of 3D subsurface
processes in the ACME Land Model also allows for a wide range of analyses heretofore
impossible in an Earth System Model context.



## 5   Tables

**Table 1. Bias, root mean square error (RMSE), and correlation ($R^2$) between modeled and observed snow depth at polygon center, rim and difference between center and rim for 2013 for three cases: Snow redistribution (SR) off and 1D physics, SR on and 1D physics, and SR on and 2D physics.**

|  | SR=Off, Physics=1D | | | SR=On, Physics=1D | | | SR=On, Physics=2D | | |
|---|---|---|---|---|---|---|---|---|---|
|  | Center | Rim | Center-Rim | Center | Rim | Center-Rim | Center | Rim | Center-Rim |
| Bias | -0.08 | 0.02 | -0.1 | -0.04 | -0.03 | -0.02 | -0.04 | -0.03 | -0.02 |
| RMSE | 0.12 | 0.04 | 0.12 | 0. 08 | 0.04 | 0.05 | 0. 08 | 0.04 | 0.05 |
| $R^2$ | 0.86 | 0.92 | 0.03 | 0.78 | 0.85 | 0.73 | 0.79 | 0.85 | 0.73 |



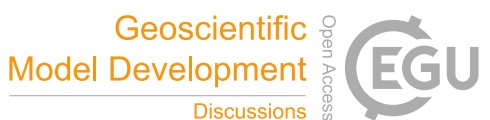

**Table 2 Bias, root mean square error (RMSE) and correlation ($R^2$) between modeled and**
**observed soil temperature at polygon center and rim at multiple soil depth for 2013 for**
**three cases: Snow redistribution (SR) off and 1D physics, SR on and 1D physics, and SR on**
**and 2D physics.**

| Bias | | | | | | |
|---|---|---|---|---|---|---|
| | SR=Off, Physics=1D | | SR=On, Physics=2D | | SR=On, Physics=2D | |
| Depth [m] | Center | Rim | Center | Rim | Center | Rim |
| 0.00 - 0.20 | 0.86 | -1.73 | -0.19 | 1.00 | 0.52 | 0.71 |
| 0.20 - 0.50 | 0.68 | -1.52 | -0.46 | 0.98 | 0.35 | 0.62 |
| 0.50 - 0.75 | 0.53 | -1.49 | -0.64 | 0.94 | 0.21 | 0.53 |
| 0.75 - 1.00 | 0.49 | -1.44 | -0.67 | -0.97 | 0.22 | 0.49 |
| Average across four depths | 0.64 | -1.54 | -0.49 | 0.97 | 0.33 | 0.59 |


| RMSE | | | | | | |
|---|---|---|---|---|---|---|
| | SR=Off, Physics=1D | | SR=On, Physics=2D | | SR=On, Physics=2D | |
| Depth [m] | Center | Rim | Center | Rim | Center | Rim |
| 0.00 - 0.20 | 2.11 | 3.39 | 2.20 | 2.94 | 1.90 | 2.66 |
| 0.20 - 0.50 | 1.49 | 2.73 | 1.39 | 1.86 | 1.12 | 1.57 |
| 0.50 - 0.75 | 1.60 | 2.42 | 1.22 | 1.96 | 1.14 | 1.60 |
| 0.75 - 1.00 | 1.50 | 2.15 | 1.12 | 1.87 | 1.09 | 1.44 |
| Average across four depths | 1.67 | 2.67 | 1.44 | 2.16 | 1.31 | 1.82 |


| $R^2$ | | | | | | |
|---|---|---|---|---|---|---|
| | SR=Off, Physics=1D | | SR=On, Physics=2D | | SR=On, Physics=2D | |
| Depth [m] | Center | Rim | Center | Rim | Center | Rim |
| 0.00 - 0.20 | 0.98 | 0.95 | 0.97 | 0.97 | 0.98 | 0.97 |



| 0.20 - 0.50 | 0.99 | 0.96 | 0.98 | 0.99 | 0.99 | 0.99 |
|---|---|---|---|---|---|---|
| 0.50 - 0.75 | 0.99 | 0.97 | 0.99 | 0.99 | 1.00 | 0.99 |
| 0.75 - 1.00 | 0.99 | 0.97 | 0.99 | 0.99 | 1.00 | 0.99 |
| Average across four depths | 0.99 | 0.96 | 0.98 | 0.99 | 0.99 | 0.99 |


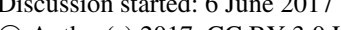


**6    Figures**

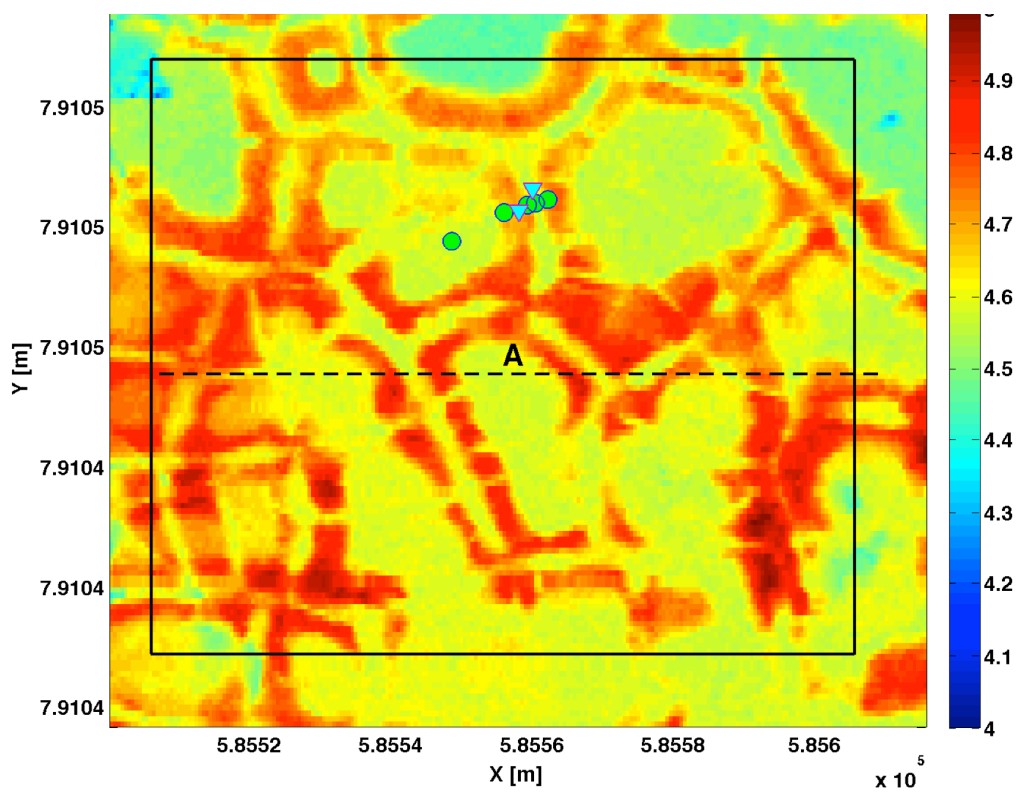


**Figure 1 The NGEE-Arctic study area A, which characterized as a low-centered polygon**
**field. Dotted line indicate the transect along which simulation in this paper are preformed**
**to demonstrate the effects of snow redistribution on soil temperature. The locations where**
**snow and temperature sensors are installed within the study site are denoted by triangle**
**and circle, respectively.**



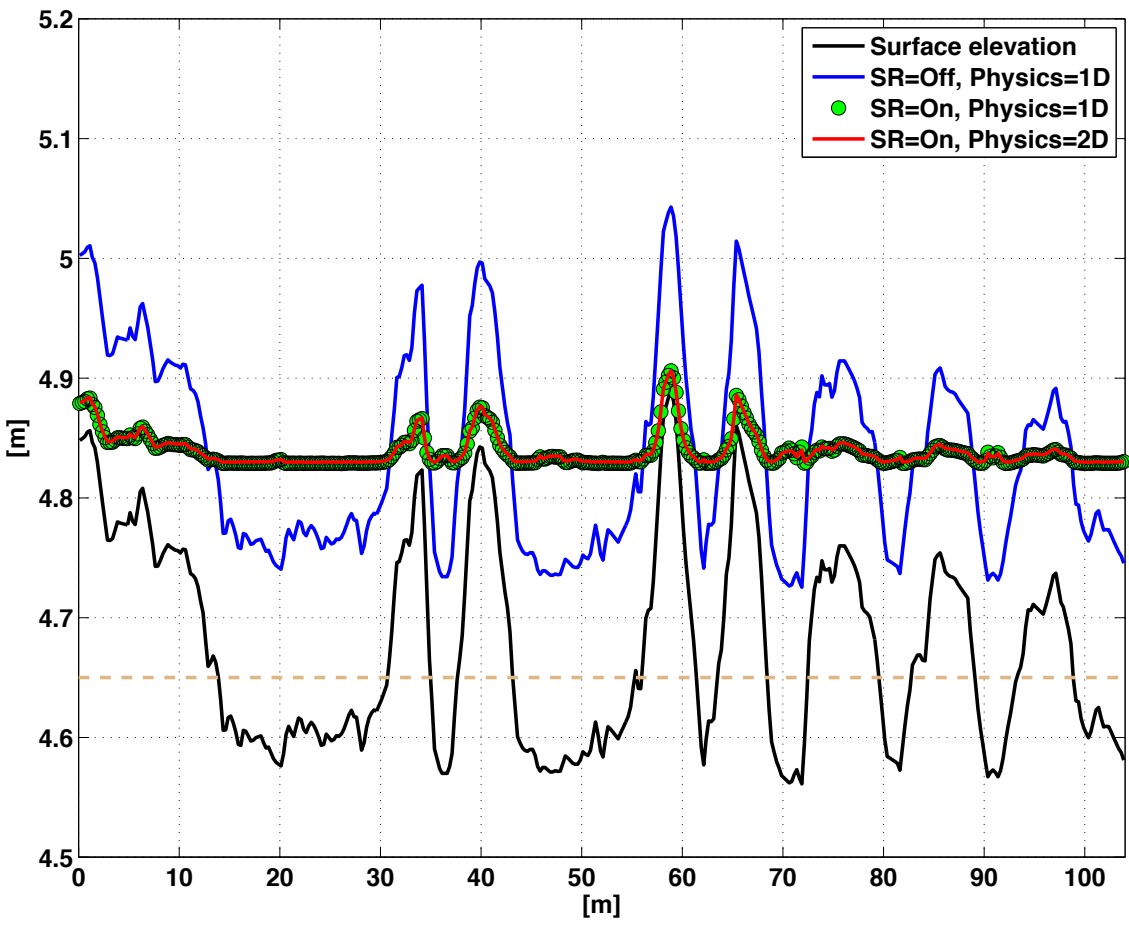


**Figure 2. Simulated average winter snow surface elevation across the transect for three scenarios: (1) snow redistribution (SR) turned off and 1D subsurface physics, (2) snow redistribution turned on and 1D subsurface physics, and (3) snow redistribution turned on and 2D subsurface physics. Surface elevation of the transect is shown by solid black line. The dashed line indicates the boundary for comparison to observations in relatively lower (centers) and relatively higher (rims) topographical positions.**





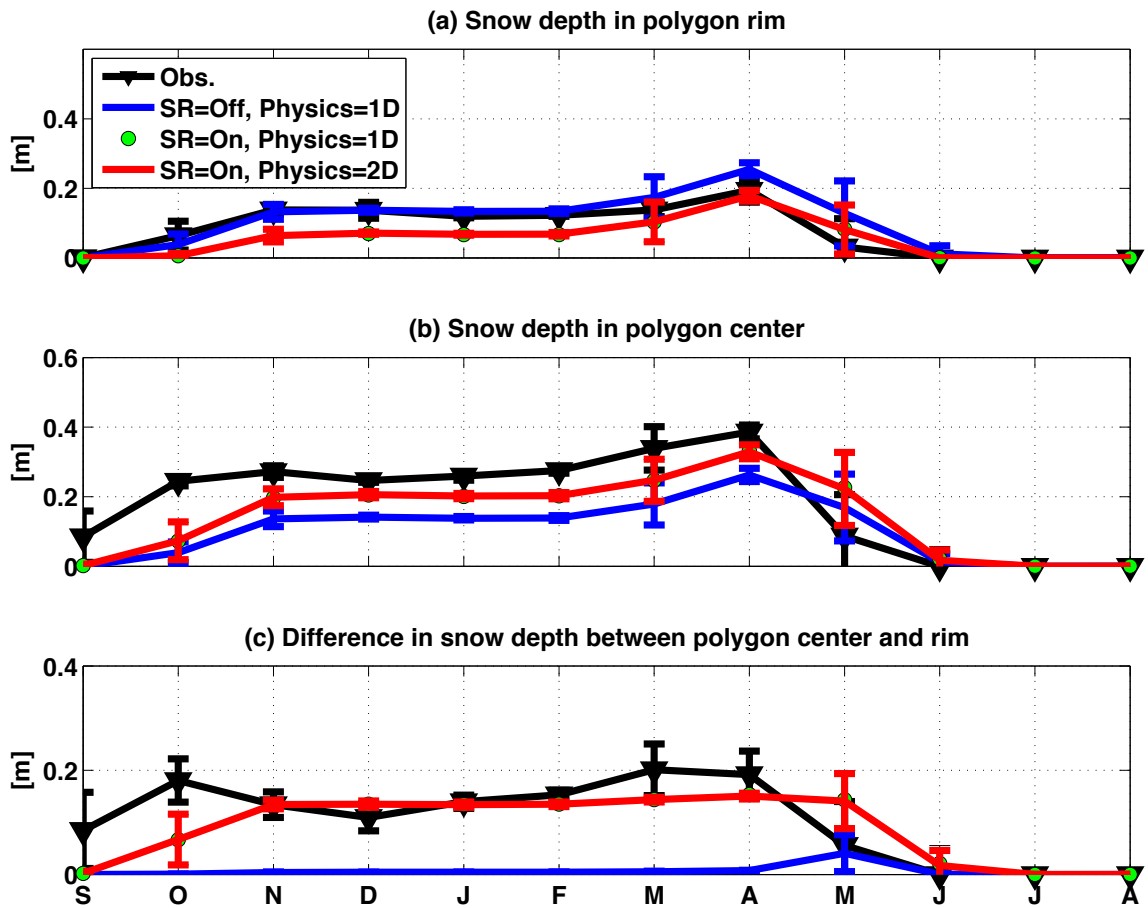



**Figure 3 Monthly-mean comparison of observation and simulated snow depth (a) in**
**polygon rim, (b) in polygon center; (c) difference between polygon center and rim for 2013.**




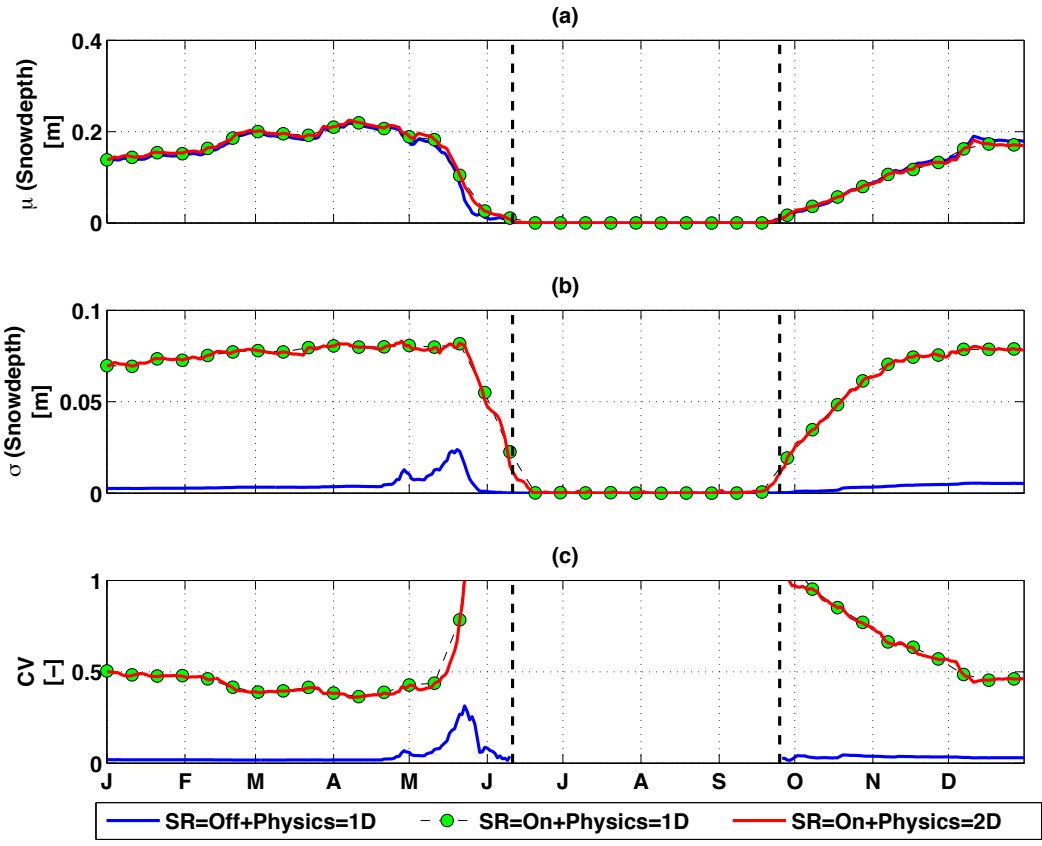


**Figure 4. Mean, standard deviation and coefficient of variation of simulated snow depth across the entire domain for 1D and 2D subsurface physics.**



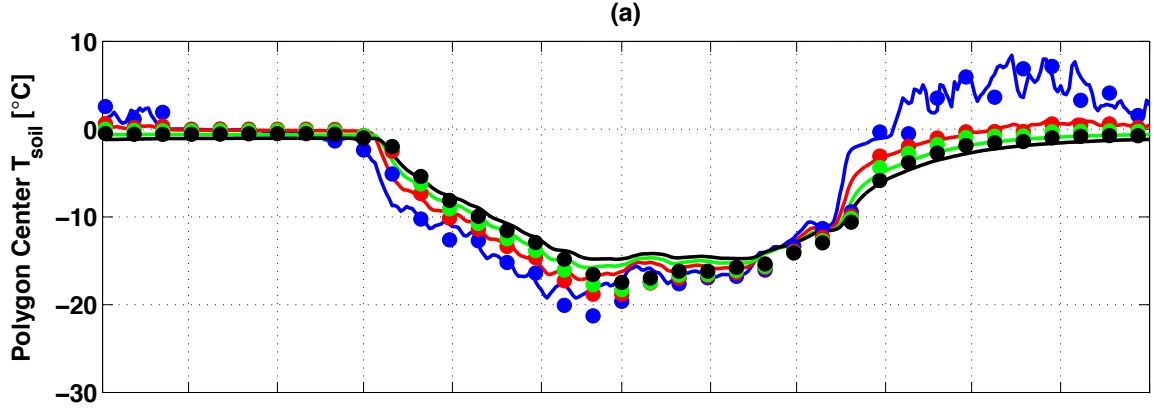

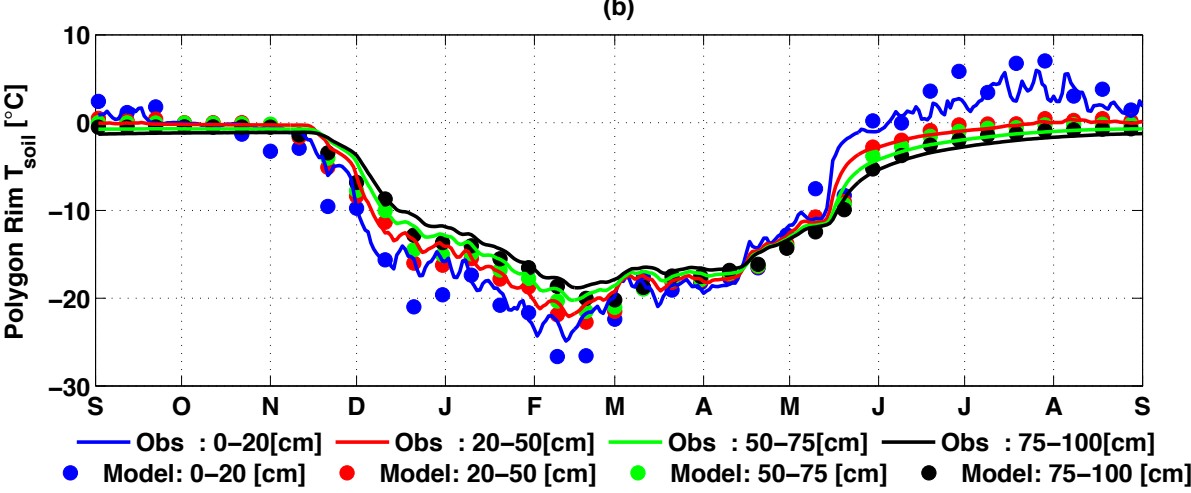


**Figure 5 Comparison of soil temperature observations and predictions in polygon centers**

**(a) and rims (b). Simulation was performed with snow redistribution on and 2D subsurface**

**physics, between September 2012 and September 2013. Simulation results are shown at an**

**interval of 10 days, while observations are shown at daily interval**

558

559



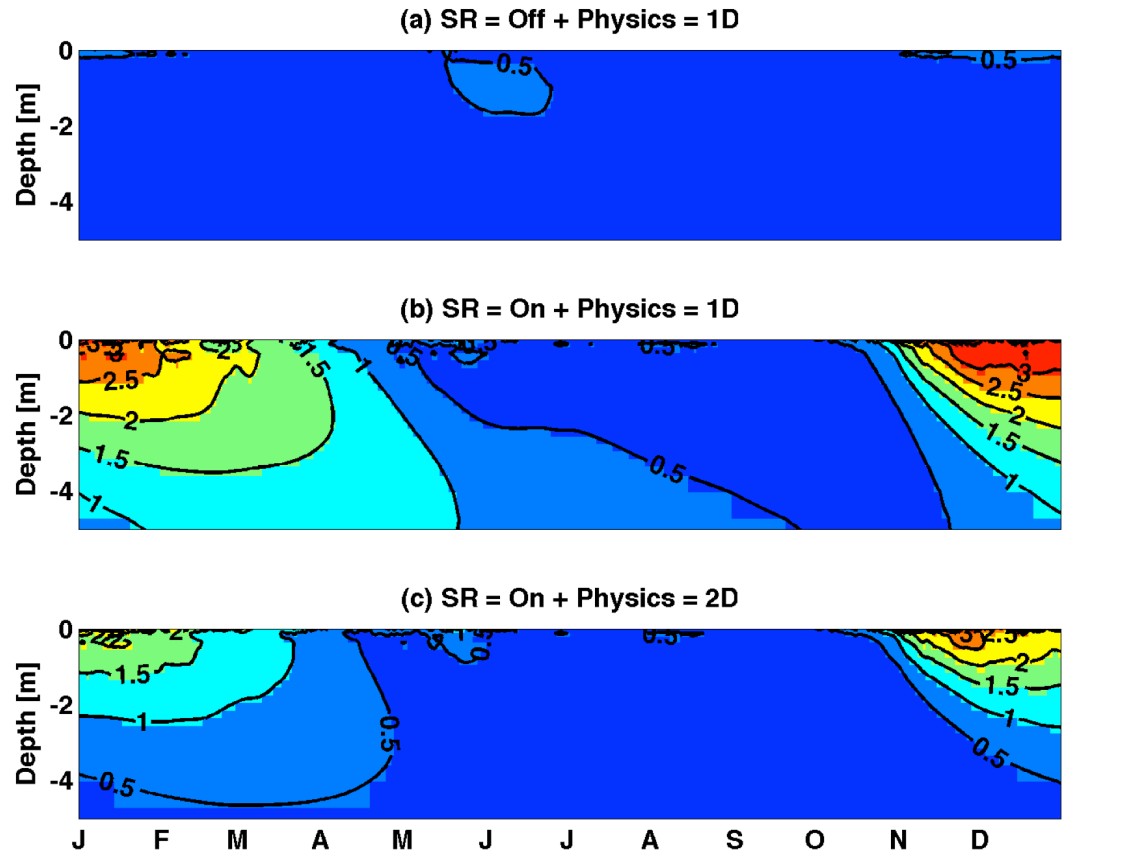

560

**Figure 6 Simulated daily spatial standard deviation averaged across 10-year of near**

**surface soil temperature for simulation performed with snow redistribution turned off and**

**1D subsurface physics (top panel); snow redistribution turned on and 1D subsurface**

**physics (middle panel); and snow redistribution turned on and 2D subsurface physics**

**(bottom panel).**







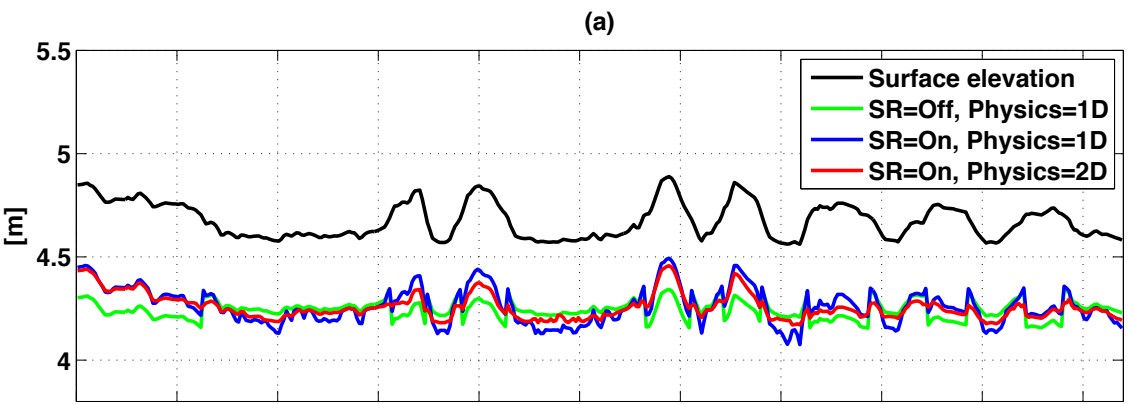

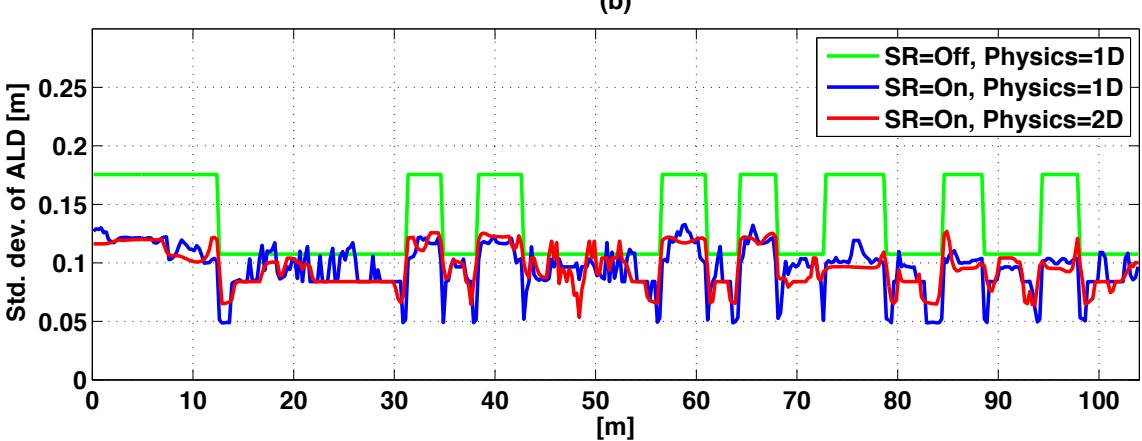



**Figure 7 Temporal mean of the bottom of the active layer (top panel) and standard**
**deviation of the active layer depth (bottom panel) over the 10-year period across the**
**modeling domain.**





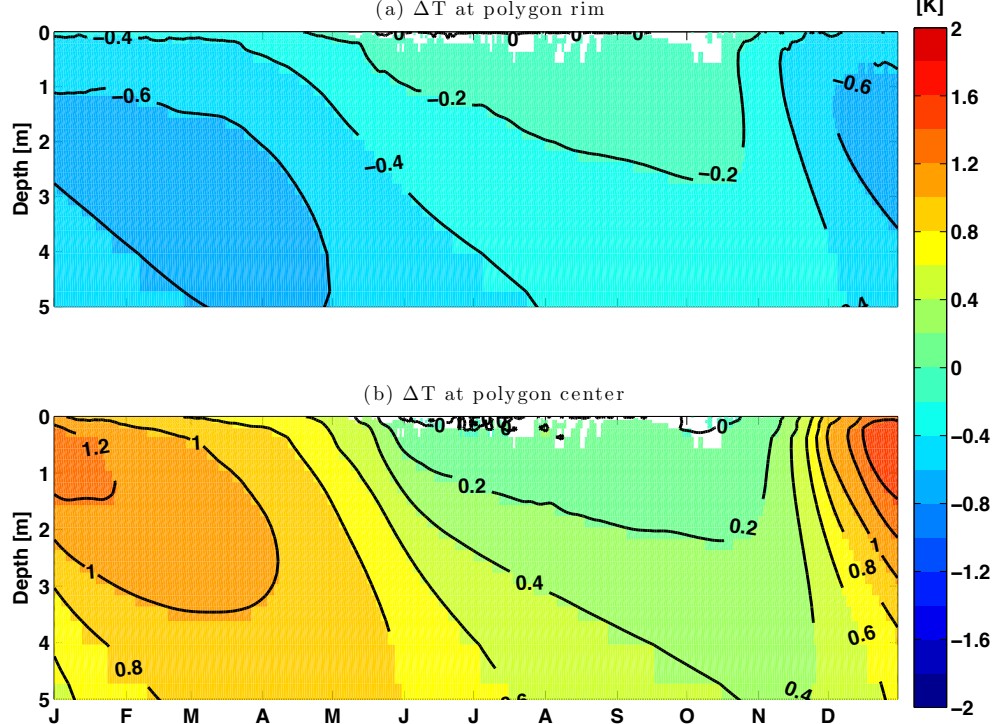


**Figure 8 Time series of spatial mean soil temperature differences between "SR=On +**

**Physics=1D" and "SR=On + Physics=2D" at polygon rim (top panel) and polygon center**

**(bottom panel).**





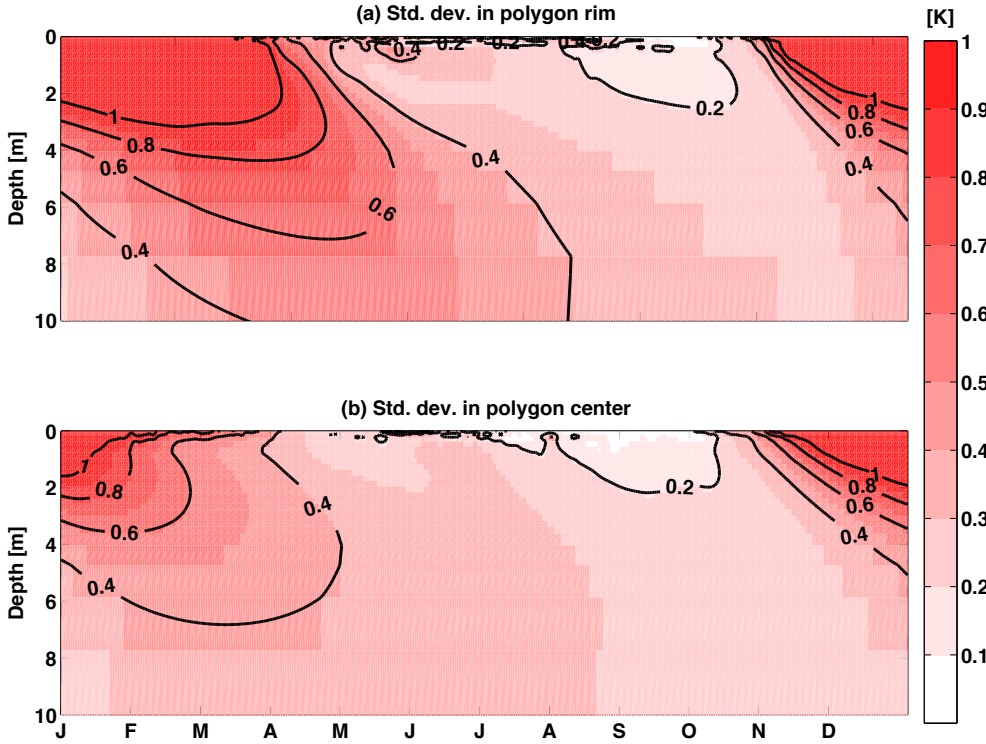


**Figure 9 Time series of soil temperature spatial standard deviation for "SR=On +**

**Physics=2D" at polygon rim (top panel) and polygon center (bottom panel).**





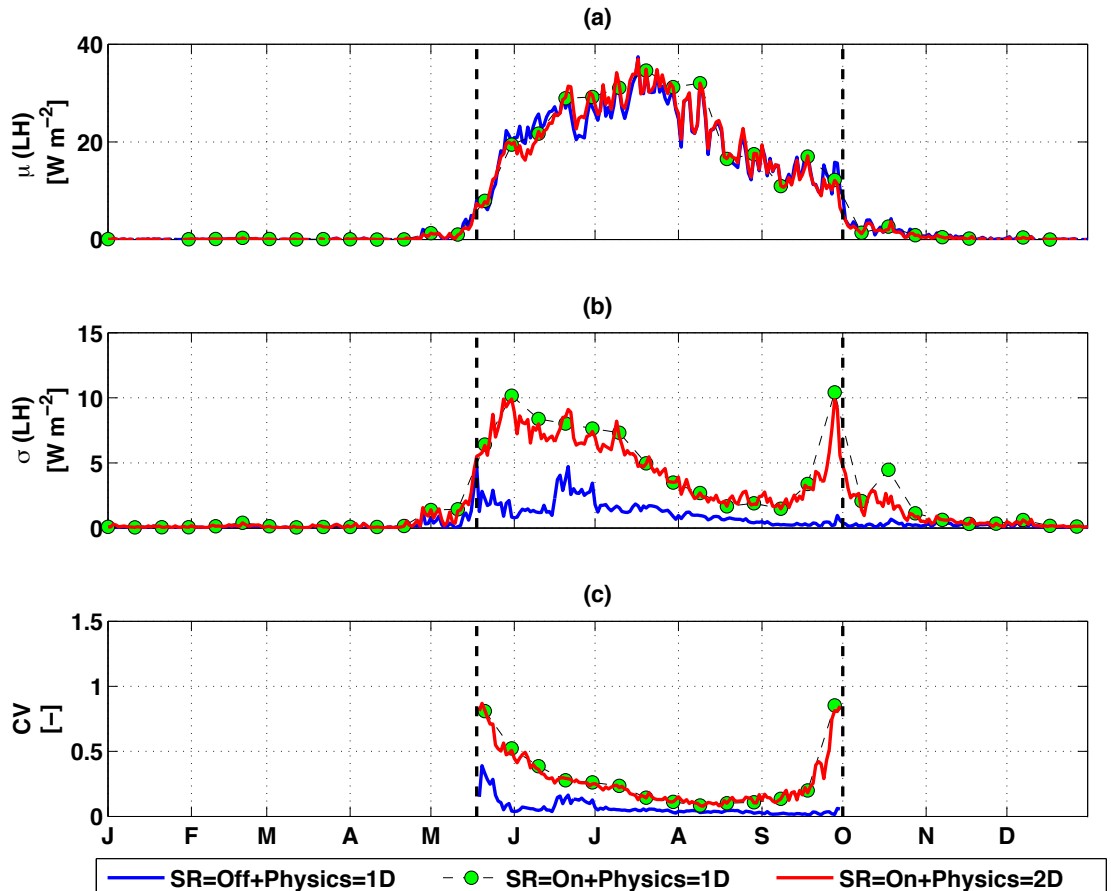



**Figure 10. Latent heat flux inter-annual (a) mean, (b) standard deviation, and (c)**
**coefficient of variation across the site A transect.**





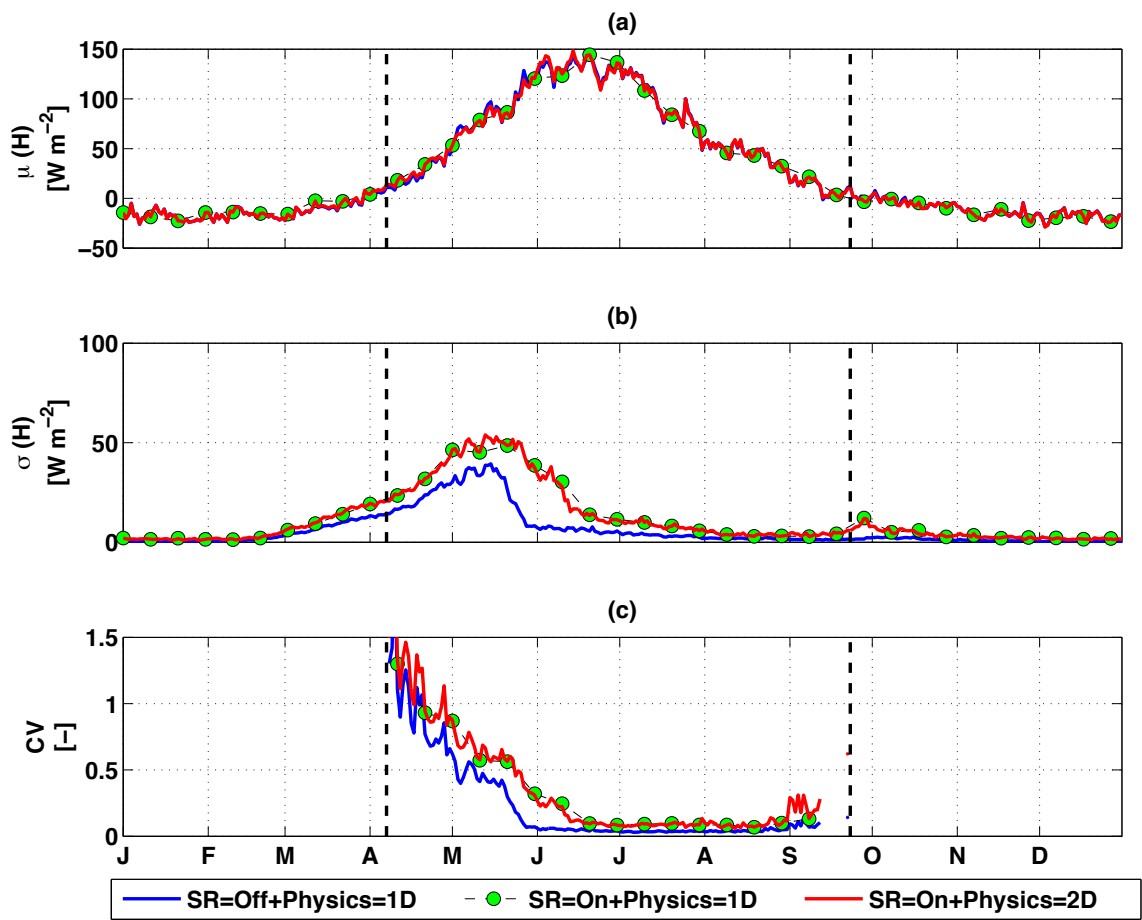



**Figure 11. Same as Figure 10 except for sensible heat flux.**





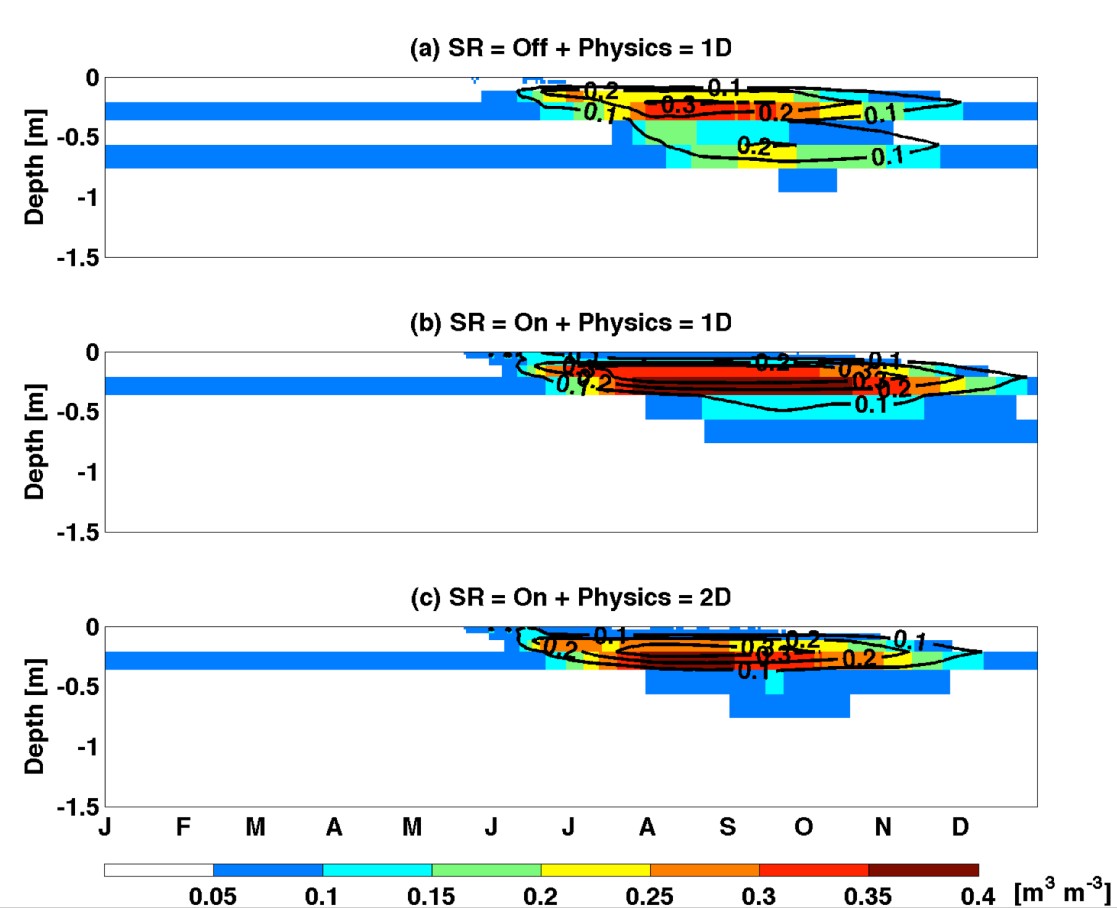


**Figure 12. Same as Figure 6 except for liquid saturation.**



**Acknowledgements***.*
This research was supported by the Director, Office of Science, Office of Biological and
Environmental Research of the US Department of Energy under Contract No. DE-AC02-
05CH11231 as part of the NGEE-Arctic and Accelerated Climate Modeling for Energy (ACME)
programs.





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
