# Peer review of "Impacts of microtopographic snow-redistribution and lateral subsurface processes on hydrologic and thermal states in an Arctic polygonal ground ecosystem"

_Geoscientific Model Development, 2017_

## Short Comment (SC1) · 8 Jun 2017

Dear authors,

In my role as Executive editor of GMD, I would like to bring to your attention our Editorial version 1.1:

http://www.geosci-model-dev.net/8/3487/2015/gmd-8-3487-2015.html

This highlights some requirements of papers published in GMD, which is also available

on the GMD website in the 'Manuscript Types' section:

http://www.geoscientific-model-development.net/submission/manuscript_types.html

In particular, please note that for your paper, the following requirements have not been met in the Discussions paper:

- "The main paper must give the model name and version number (or other unique identifier) in the title."

- "If the model development relates to a single model then the model name and the version number must be included in the title of the paper. If the main intention of an article is to make a general (i.e. model independent) statement about the usefulness of a new development, but the usefulness is shown with the help of one specific model, the model name and version number must be stated in the title. The title could have a form such as, "Title outlining amazing generic advance: a case study with Model XXX (version Y)"."

- "All papers must include a section, at the end of the paper, entitled 'Code availability'. Here, either instructions for obtaining the code, or the reasons why the code is not available should be clearly stated. It is preferred for the code to be uploaded as a supplement or to be made available at a data repository with an associated DOI (digital object identifier) for the exact model version described in the paper. Alternatively, for established models, there may be an existing means of accessing the code through a particular system. In this case, there must exist a means of permanently accessing the precise model version described in the paper. In some cases, authors may prefer to put models on their own website, or to act as a point of contact for obtaining the code. Given the impermanence of websites and email addresses, this is not encouraged, and authors should consider improving the availability with a more permanent arrangement. After the paper is accepted the model archive should be updated to include a link to the

GMD paper."

- Inclusion of Code and/or data availability sections is mandatory for all papers and should be located at the end of the article, after the conclusions, and before any appendices or acknowledgments. For more details refer to the code and data policy.

Please correct the title and add a code availability section.

Yours,

Astrid Kerkweg

---

## Referee Comment (RC1) · Anonymous Referee #1 · 7 Jul 2017

**General comments:**

Manuscript by Bisht et al. presents simulation results in an Arctic polygonal ground ecosystem using an improved ALM model including lateral processes and snow redistribution. The conclusions are partly supported by modeling results, e.g., 1) snow depth variation was affected by snow redistribution, but not by lateral processes of thermal flow, 2) active layer depths was affected by lateral energy fluxes. Like many others, this work again stresses that advances in the land surface modeling is needed. In fact, the simple snow redistribution approach in the paper can be readily

incorporated into land models. My main reservations are the selection of the 2D transect and model validation. Why the transect is not selected where the sensors (as shown In Figure 1) are located? It makes the comparison between the model and observation meaningless.

**Specific comments:**
1) Lengthy texts in the Introduction that are not directly related to the study.
2) Line 100-101: define "active layer thickness" for general readers.
3) Line 126: define ALM.
4) Line 158-160: redundant as already described in lines 126-128.
5) Line 169: check unit of Q.
6) Define z in Eq. 2 and other variables in Eq. 4.
7) Eqs. 17 and 18, check the third term on the RHS.
8) Eq. 23: write $c_n$ as $c_{i,j,k}$
9) Define $\omega$' in Eqs. 25-31
10) Line 312: from Fig. 2, I see less dependence of average snow depth on topography with SR.
11) How well is the 3D model developed in the paper compared to analytical solutions or other well established numerical models?
12) Where are the locations of center and rim in the model simulations? Fig. 1 shows two snow sensors and five temperature sensors. At what locations are the simulation compared to the corresponding observations?
13) As the authors noted on line 246 that PETSc is a scalable solver, so what is constraining the 3D simulation (statement on line 447)?
14) Because of the computational constraint, I don't agree with the last statement on line 510-512.
15) Figure 1: what's the legend? DEM?

---

## Referee Comment (RC2) · Anonymous Referee #2 · 13 Jul 2017

This is a well-written paper that described in a very clear and clean manner small-scale model simulations of the impact of snow redistribution and lateral subsurface processes. The model is set up on a well instrumented study site and the results are comprehensively analyzed and honestly discussed. The results are clear and allow for an unambiguous prioritization of effects to take into account in lager-scale simulations (snow redistribution seems to be much much more of a pressing issue than 2D or 3D soil physics). This paper clearly deserves being published, but I have some concerns about the modeling strategy in general that should be addressed in a revised version.

[Figure]

I also have some specific remarks and comments.

General remark. The framework of the paper is Earth System modeling. The authors implement small-scale snow redistribution and 3D soil physics (2D in the setup used here). The results show that a simple snow redistribution parameterization based on microtopography has a very beneficial effect on a range of simulated variables. This is very nice. However, I think that the paper almost entirely misses a thorough discussion of an implementation strategy for these development in the ultimate context of Earth System modeling. This will happen on much larger spatial scales. How will you move from an explicit fine-scale representation to a sub grid implementation? Will the choice be only to include snow redistribution (i.e. aren't there already enough results to decide that a 3D soil physics will be an ÂńÂăoverkillÂăÂż in the Earth System modeling context)? Will the model have two tiles (polygon centers and rims), with snow being shuffled from one tile to the other? Or is the whole thing probably going to be more complex, with an explicit modeling of 3D soil physics supposing an idealized polygon of some finite size? What will be done if the model domain does include areas that are not polygonal tundra (it's supposed to be a global model if I understand correctly)? If there are issues with computing time already in a 2d setting, is it realistic to go to 3d? Some words on validation/tests on larger scales? Answers to some of these questions might be pretty obvious, but I nevertheless think that a proper discussion of these and other related questions is required.

Specific comments. - L.24 : "Three ten-years long simulations" : Is that good English?

- L.55 : "Xu, 2016#154"

- L61: The reference to Friedlingstein et al., 2006 is good but there has been quite some work on this more recently. In general, there are very many pre-2007 references and much less after that period. Maybe the bibliography could be a bit updated. For example, in line 78, the review by Schuur et al. in Nature 2015 might be worth citing.

- L.166. "The flow water" -> "The water flow" or "The flow of water"

- L. 198. I suggest to clarify the writing here. What about this: ". . ... zeta is the diagonal entry of the banded matrix (eq. 11-17)", then provide eq. 11-17. Then: "small phi is a column vector given by:", then put eq. 18. I think that would be clearer.

- The same applies to eqs. 25-32. Separate eq. 32 from 25-31. I think that eq. 28 should read "eta=..." (not "mu=...") and eq. 29 should read "mu=..." (not "xi=...")

- Line 232: Please say clearly that this means that there is no geothermal heat flux represented in the model.

- L. 261: "to simulate SR", not "to simulated SR"

- L. 273: "its", not "it's"

- L.277: A broken link to some internal reference. same at line 328, 342, 343

- L. 285: with do you put the dimension meters in square brackets?

- L. 289: "SP mode": that's an internal nickname. Its meaning becomes clear at the end of the paper ("satellite phenology") but this is not required here. Either explain the acronym of leave it out.

---

## Author Comment (AC2) · 23 Aug 2017

**Impacts of microtopographic snow-redistribution and lateral subsurface processes**
**on hydrologic and thermal states in an Arctic polygonal ground ecosystem [MS no.**
**gmd-2017-71]**

**RC1: 'Review of the manuscript by Bisht et al.', Anonymous Referee #1**

General comments:

Manuscript by Bisht et al. presents simulation results in an Arctic polygonal ground
ecosystem using an improved ALM model including lateral processes and snow
redistribution. The conclusions are partly supported by modeling results, e.g., 1) snow
depth variation was affected by snow redistribution, but not by lateral processes of thermal
flow, 2) active layer depths was affected by lateral energy fluxes. Like many others, this
work again stresses that advances in the land surface modeling is needed. In fact, the
simple snow redistribution approach in the paper can be readily incorporated into land
models.

*My main reservations are the selection of the 2D transect and model validation. Why the*
*transect is not selected where the sensors (as shown In Figure 1) are located? It makes the*
*comparison between the model and observation meaningless.*

**Response:**

We acknowledge that the 2D transect used for simulations in this study does not align with
the sensor location. The objective of this work was not to validate the model for the few
grid cells that exactly align with the observations recorded in the rim and center of a
polygon, but to quantify relative differences between simulations for rim and center of a
polygon. As noted in Figure 2, all grid cells above the dashed line were classified as rim,
while all grid cells below the dashed line were classified as center. The model accurately
captures the snow depth differences between rim and center when SR is turned on (Table
1). Additionally, errors in simulated temperature for all soil depths are lower for rim and
center when SR is included (Table 2). Thus, our comparison of model results against
observations is reasonable and the comparison we present indicates the model accurately
represents system characteristics important for the conclusions of our paper.

Specific comments:

*1) Lengthy texts in the Introduction that are not directly related to the study.*

**Response:**

We have removed text in introduction describing changes in NEP within Arctic ecosystems as simulation in this work did not have an active biogeochemistry cycle.

*2) Line 100-101: define "active layer thickness" for general readers.*

**Response:**

We have added a definition for active layer thickness.

*3) Line 126: define ALM.*

**Response:**

We have updated the text to define ALM.

*4) Line 158-160: redundant as already described in lines 126-128.*

**Response:**

We have updated the text to remove redundancy.

*5) Line 169: check unit of Q.*

**Response:**

The units of $Q$ have been corrected to [m$^{-3}$ of water m$^{-3}$ of soil s$^{-1}$]

*6) Define z in Eq. 2 and other variables in Eq. 4.*

**Response:**

All terms in Equation 2 and 4 are now defined.

*7) Eqs. 17 and 18, check the third term on the RHS.*

**Response:**

Third term in equation 17 and 18 is updated.

*8) Eq. 23: write cn as ci,j,k*

**Response:**

In equation 23, $c_n$ is now defined as $c_{n_{i,j,k}}$. Additionally, equations 25-32 have been updated.

*9) Define ω' in Eqs. 25-31*

**Response:**

In equation 25-31, $\omega'$ is now replaced by $1 - \omega$, where $\omega$ is defined as the weight in the

Crank-Nicholson method.

*10) Line 312: from Fig. 2, I see less dependence of average snow depth on topography with*

*SR.*

**Response:**

We have fixed the typographical error and the text now reads "*With SR, a much smaller*

*dependence of winter-average snow depth on topography is predicted*"

*11) How well is the 3D model developed in the paper compared to analytical solutions or*

*other well established numerical models?*

**Response:**

In this work, we extended the existing 1D physics formulations for subsurface hydrologic and thermal processes to included lateral processes. Thus, we did not compare existing physics formulations against analytical solutions or other numerical models, but we did ensure that lateral coupling was implemented correctly. Sanity checks were preformed to ensure the 3D model solution is the same as in the 1D vertical model when the problem setup is horizontally homogeneous (Results not shown).

The thermal model is independent of gravity. Thus, additional tests were performed to ensure the numerical solution of the thermal model for propagation of heat is identical in a 1D column that is oriented horizontally and vertically. A test was performed to study the propagation of a heat perturbation that was applied on the left and top boundary of a spatially homogeneous 2D domain (Figure 1, below). The difference of simulated temperature between the two cases was of the order of the tolerance of the numerical solver (Figure 1c). An additional test was performed in which a sinusodially varying temperature perturbation was applied on the left and top boundary; and the difference in results was again within tolerance of numerical solver (Figure 2). These tests ensured that lateral coupling was correctly implemented within the model. To address the reviewer's concerns regarding testing, we have added description of these analyses to the

Supplementary Material (Page 2, lines 18-40, and a reference to these tests has been added to the main text (Page 12, lines 241-244).

[Figure]

**Figure 1. Propagation of a spatially homogeneous temperature perturbation applied**

**on the (a) left and (b) top boundary of a spatially homogeneous 2D transect at the**

**end of 1-day. (c) The difference in evolved temperature between two cases is many**

**orders of magnitude smaller than the predicted states.**

[Figure]

**Figure 2 Same as Figure 1 except a sinusoidally varying spatial temperature perturbation is applied.**

*12) Where are the locations of center and rim in the model simulations? Fig. 1 shows two snow sensors and five temperature sensors. At what locations are the simulation compared to the corresponding observations?*

**Response:**

The dashed line in Figure 2 classifies the 2D transect into rim and center. All grid cells that have surface elevation above the dash line are classified as rim, while all grid cells below the dashed line are marked as center.

*13) As the authors noted on line 246 that PETSc is a scalable solver, so what is constraining the 3D simulation (statement on line 447)?*

**Response:**

ALM is embarrassing parallel and has no cross processor communication because it is a 1D, vertical-only model. Even though PETSc is a scalable solver, the current implementation of the 3D model is serial. Thus, our model is capable of solving a 3D problem on each processor independently but unable to solve a parallel, 3D problem. We have updated the text in Section 3.5 (Page 19, lines 443-447) to clarify this point.

*14) Because of the computational constraint, I don't agree with the last statement on line 510-512.*

**Response:**

We have updated the text to reflect that the current model is serial (Page 19, Lines 444-445). Even though the current version of the ALM-3D model is sequential, we believe it would be very useful for applications in the Earth System Model context. One potential future application would be to solve 3D subsurface hydrologic and thermal processes within a watershed. To this end, the domain decomposition of ALM in future versions could be modified such that all grid cells within a watershed are assigned to a single processor. In such an application, ALM-3D v1 would be an appropriate candidate.

*15) Figure 1: what's the legend? DEM?*

**Response:**

The legend indicates the height in meters (now added to Figure 1).

---

## Author Comment (AC3) · 23 Aug 2017

**Impacts of microtopographic snow-redistribution and lateral subsurface processes**

**on hydrologic and thermal states in an Arctic polygonal ground ecosystem [MS no.**

**gmd-2017-71]**

**RC2: 'A useful contribution', Anonymous Referee #2**

General remark. The framework of the paper is Earth System modeling. The authors implement small-scale snow redistribution and 3D soil physics (2D in the setup used here).

The results show that a simple snow redistribution parameterization based on microtopography has a very beneficial effect on a range of simulated variables. This is very nice. However, I think that the paper almost entirely misses a thorough discussion of an implementation strategy for these development in the ultimate context of Earth System modeling. This will happen on much larger spatial scales.

How will you move from an explicit fine-scale representation to a sub grid implementation?

Will the choice be only to include snow redistribution (i.e. aren't there already enough results to decide that a 3D soil physics will be an overkill in the Earth System modeling context)? Will the model have two tiles (polygon centers and rims), with snow being shuffled from one tile to the other? Or is the whole thing probably going to be more complex, with an explicit modeling of 3D soil physics supposing an idealized polygon of some finite size? What will be done if the model domain does include areas that are not polygonal tundra (it's supposed to be a global model if I understand correctly)?

**Response:**

This study is a necessary first step of documenting the role of fine scale processes associated with microtopography and lateral redistribution of water and energy in the subsurface. We acknowledge that a development of a sub grid structure to parsimoniously capture impacts of microtopography and lateral subsurface processes on coarser grid scale is a worthy scientific research, but such a new development is beyond the scope of the current work.

However, here are some thoughts on possible approaches to parsimoniously include fine scale processes. As suggested by the reviewer, investigate how accurate is a two-tile approach as compared to explicitly modeling the transect when snow redistribution is
accounted for within the model. Additional simulations will be needed to investigate how
well the two-tile approach performs when biogeochemical cycling is included. Exclusion of
lateral subsurface processes has a greater impact on predicted subgrid variability than on
spatially averaged states. Thus, one possible extension of the current model would be to
explicitly include an equation for the temporal evolution of sub grid variability of using the
approach of Montaldo and Albertson (2003). The use of reduced-order models as described
by Pau et al. (2014) is an alternate approach to estimate fine scale hydrologic and thermal
states from coarse resolution simulation. We have added discussion of these topics to the
Discussion section (page 20, Lines 468-4477)
If there are issues with computing time already in a 2d setting, is it realistic to go to 3d?
**Response:**
Moving beyond a 1D land model to a 2D/3D model will certainly increase the
computational cost of the simulation. However, the land component is typically the least
expensive component of an Earth System Model. ALM is less than 5% of the total
computational cost of a fully coupled ACME simulation (ACME Performance team, personal
communication, May 25, 2017). Even though there is some leeway in increasing the
computational cost of the land model, the need to include higher spatial dimensional
processes in land surface models has been made by many studies (Chen et al. (2006); Kim
and Mohanty (2016); Maxwell and Condon (2016)). Lateral subsurface processes can be
included in the land surface model via a range of numerical discretization approaches of
varying complexity such as adding lateral flux of water and energy as source/sink term in
the existing 1D model, implementing an operator split approach to solve vertical and
lateral processes in a non-iterative model, or solving a fully coupled 3D model. Increased
computational cost is not the only factor limiting application of ALM-3D to a global
simulation. The subgrid hierarchy structure of the land model, which presently does not
have any topological information, needs to be updated to include lateral connectivity. We
have added some Discussion on theses topics to the revised version (Page 20, Lines 477-
483).

Some words on validation/tests on larger scales?

**Response:**

Model validation is an integral part of model development. Ongoing projects of the U.S

Department of Energy such as the NGEE-Arctic (https://ngee-arctic.ornl.gov) and the

NGEE-Tropics ([http://ngee-tropics.lbl.gov/](http://ngee-tropics.lbl.gov/)) are expected to provide a wide range datasets related to land surface model at regional scales. Additionally, the Distributed Model

Intercomparison Project Phase 2 (DMIP 2) provides a comprehensive datasets and modeling protocol for benchmarking distributed hydrologic models (Smith et al., 2012) and estimates of water table depth at global scales are available from Fan et al. (2013). Our future work will focus on application and validation of ALM-3D at regional scales. We have added some discussion of these issues to the Discussion section (page 20, Lines 483-486)

Answers to some of these questions might be pretty obvious, but I nevertheless think that a proper discussion of these and other related questions is required.

**Response:**

We added text in the discussion section that answers all of the questions raised by the reviewer.

Specific comments.

- L.24 : "Three ten-years long simulations" : Is that good English?

**Response:**

The text has been modified to "Multiple 10-years long simulations"

- L.55 : "Xu, 2016#154"

**Response:**

The incorrect citation has now been removed in the updated version of the manuscript.

- L61: The reference to Friedlingstein et al., 2006 is good but there has been quite some work on this more recently. In general, there are very many pre-2007 references and much less after that period. Maybe the bibliography could be a bit updated. For example, in line

78, the review by Schuur et al. in Nature 2015 might be worth citing.

**Response:**

- L.166. "The flow water" -> "The water flow" or "The flow of water"

**Response:**

The text has been updated to 'The flow of water'.

- L. 198. I suggest to clarify the writing here. What about this: ". . .. zeta is the diagonal entry of the banded matrix (eq. 11-17)", then provide eq. 11-17. Then: "small phi is a column vector given by:", then put eq. 18. I think that would be clearer.

**Response:**

As per reviewer suggestions, description of equations 11-18 has been separated into a description of equations 11-17 followed by a description of equation 18.

- The same applies to eqs. 25-32. Separate eq. 32 from 25-31. I think that eq. 28 should read

"eta=..." (not "mu=...") and eq. 29 should read "mu=..." (not "xi=...")

**Response:**

As per reviewer suggestion, description of equations 25-32 has been separated into two.

Additionally, equations 28 and 29 have been correctly updated.

- Line 232: Please say clearly that this means that there is no geothermal heat flux represented in the model.

**Response:**

The text updated to explicitly state that geothermal heat flux was not accounted for in this work.

- L. 261: "to simulate SR", not "to simulated SR"

**Response:**

The text has been updated.

- L. 273: "its", not "it's"

**Response:**

The text has been updated.

- L.277: A broken link to some internal reference. same at line 328, 342, 343

**Response:**

All broken references have been updated.

- L. 285: with do you put the dimension meters in square brackets?

**Response:**

Square brackets have been removed.

- L. 289: "SP mode": that's an internal nickname. Its meaning becomes clear at the end of the paper ("satellite phenology") but this is not required here. Either explain the acronym of leave it out.

**Response:**

Text has been updated to explain the acronym.

**References**

Chen, Y., Hall, A., and Liou, K. N.: Application of three-dimensional solar radiative transfer to mountains, Journal of Geophysical Research: Atmospheres, 111, n/a-n/a, 2006.

Fan, Y., Li, H., and Miguez-Macho, G.: Global Patterns of Groundwater Table Depth, Science,

339, 940-943, 2013.

Kim, J. and Mohanty, B. P.: Influence of lateral subsurface flow and connectivity on soil water storage in land surface modeling, Journal of Geophysical Research: Atmospheres,

121, 704-721, 2016.

Maxwell, R. M. and Condon, L. E.: Connections between groundwater flow and transpiration partitioning, Science, 353, 377-380, 2016.

Montaldo, N. and Albertson, J. D.: Temporal dynamics of soil moisture variability: 2.

Implications for land surface models, Water Resources Research, 39, n/a-n/a, 2003.

Pau, G. S. H., Bisht, G., and Riley, W. J.: A reduced-order modeling approach to represent subgrid-scale hydrological dynamics for land-surface simulations: application in a polygonal tundra landscape, Geosci. Model Dev., 7, 2091-2105, 2014.

Smith, M. B., Koren, V., Reed, S., Zhang, Z., Zhang, Y., Moreda, F., Cui, Z., Mizukami, N.,

Anderson, E. A., and Cosgrove, B. A.: The distributed model intercomparison project – Phase

2: Motivation and design of the Oklahoma experiments, Journal of Hydrology, 418, 3-16,

2012.

---

## Author Response (AR1)

18 August 2017

Dear Dr. Peylin,

My co-authors and I are pleased to submit our revised manuscript titled "*Impacts of microtopographic snow-redistribution and lateral subsurface processes on hydrologic and thermal states in an Arctic polygonal ground ecosystem*" for your consideration for publication in Geoscientific Model Development.

We thank you, the executive editor, and the two reviewers for insightful and constructive feedback, which helped us to calrify important aspects of our work. Modifications made in the revised version of the manuscript as compared to initial submission are summarized below:

- 1. As suggestion by reviewer #1, the introduction section has been shortened by removing the description of changes in Arctic net ecosystem productivity.
- 2. The discussion regarding future work has been expanded to include possible approaches to parsimoniously represent fine scale processes within a global land model.
- 3. We added to supplimentary information a description of numerical tests we performed to ensure new model developments were correctly implemented.
- 4. The code availability section has been revised to included reference to the publicly accessible code and dataset repositories that were used in this study.

My co-authors and I believe we have thoroughly addressed all the reviewer comments and that the revised manuscript is well suited for publication in Geoscientific Model Development. We look forward to receiving your response.

Sincerely, Gautam Bisht Impacts of microtopographic snow-redistribution and lateral subsurface processes
 on hydrologic and thermal states in an Arctic polygonal ground ecosystem [MS no.
 gmd-2017-71]

4

SC1: 'Executive Editor Comment on "Impacts of microtopographic snow redistribution and lateral subsurface processes on hydrologic and thermal states in
 an Arctic polygonal ground ecosystem"', Astrid Kerkweg

8

9 "The main paper must give the model name and version number (or other unique identifier)
10 in the title."

"If the model development relates to a single model then the model name and the version number must be included in the title of the paper. If the main intention of an article is to make a general (i.e. model independent) statement about the usefulness of a new development, but the usefulness is shown with the helpof one specific model, the model name and version number must be stated in the title. The title could have a form such as, "Title outlining amazing generic advance: a case study with Model XXX (version Y)"."

**17 Response:**

We have updated the title of our manuscript to be "Impacts of microtopographic snowredistribution and lateral subsurface processes on hydrologic and thermal states in an Arctic polygonal ground ecosystem: A case study using ALM-3D v1.0"

21

22 "All papers must include a section, at the end of the paper, entitled 'Code availability'. Here, 23 either instructions for obtaining the code, or the reasons why the code is not available should 24 be clearly stated. It is preferred for the code to be uploaded as a supplement or to be made 25 available at a data repository with an associated DOI (digital object identifier) for the exact 26 model version described in the paper. Alternatively, for established models, there may be an 27 existing means of accessing the code through a particular system. In this case, there must exist 28 a means of permanently accessing the precise model version described in the paper. In some 29 cases, authors may prefer to put models on their own website, or to act as a point of contact 30 for obtaining the code. Given the impermanence of websites and email addresses, this is not 31 encouraged, and authors should consider improving the availability with a more permanent

- 32 arrangement. After the paper is accepted the model archive should be updated to include a
- 33 link to the GMD paper."
- 34 Inclusion of Code and/or data availability sections is mandatory for all papers and should be
- 35 located at the end of the article, after the conclusions, and before any appendices or
- 36 acknowledgments. For more details refer to the code and data policy.
- 37 Response:
- 38 We have publicly released the code and data used in this study. The ALM-3D code is
- 39 available at https://bitbucket.org/gbisht/lateral-subsurface-model, while the data used in
- 40 this study is available at https://bitbucket.org/gbisht/notes-for-gmd-2017-71.

**41 RC1: 'Review of the manuscript by Bisht et al.', Anonymous Referee #1**

42

**43 General comments:**

44 Manuscript by Bisht et al. presents simulation results in an Arctic polygonal ground 45 ecosystem using an improved ALM model including lateral processes and snow 46 redistribution. The conclusions are partly supported by modeling results, e.g., 1) snow 47 depth variation was affected by snow redistribution, but not by lateral processes of thermal 48 flow, 2) active layer depths was affected by lateral energy fluxes. Like many others, this 49 work again stresses that advances in the land surface modeling is needed. In fact, the 50 simple snow redistribution approach in the paper can be readily incorporated into land 51 models.

52

53 My main reservations are the selection of the 2D transect and model validation. Why the 54 transect is not selected where the sensors (as shown In Figure 1) are located? It makes the 55 comparison between the model and observation meaningless.

**56 **Response**:**

57 We acknowledge that the 2D transect used for simulations in this study does not align with 58 the sensor location. The objective of this work was not to validate the model for the few 59 grid cells that exactly align with the observations recorded in the rim and center of a polygon, but to quantify relative differences between simulations for rim and center of a 60 polygon. As noted in Figure 2, all grid cells above the dashed line were classified as rim, 61 62 while all grid cells below the dashed line were classified as center. The model accurately 63 captures the snow depth differences between rim and center when SR is turned on (Table 1). Additionally, errors in simulated temperature for all soil depths are lower for rim and 64 65 center when SR is included (Table 2). Thus, our comparison of model results against observations is reasonable and the comparison we present indicates the model accurately 66 67 represents system characteristics important for the conclusions of our paper.

68

69 Specific comments:

1) Lengthy texts in the Introduction that are not directly related to the study.

71 **Response:**

| 72 | We have removed text in introduction describing changes in NEP within Arctic ecosystems             |
|----|-----------------------------------------------------------------------------------------------------|
| 73 | as simulation in this work did not have an active biogeochemistry cycle.                            |
| 74 |                                                                                                     |
| 75 | 2) Line 100-101: define "active layer thickness" for general readers.                               |
| 76 | Response:                                                                                           |
| 77 | We have added a definition for active layer thickness.                                              |
| 78 |                                                                                                     |
| 79 | 3) Line 126: define ALM.                                                                            |
| 80 | Response:                                                                                           |
| 81 | We have updated the text to define ALM.                                                             |
| 82 |                                                                                                     |
| 83 | 4) Line 158-160: redundant as already described in lines 126-128.                                   |
| 84 | Response:                                                                                           |
| 85 | We have updated the text to remove redundancy.                                                      |
| 86 |                                                                                                     |
| 87 | 5) Line 169: check unit of Q.                                                                       |
| 88 | Response:                                                                                           |
| 89 | The units of $Q$ have been corrected to $[m^{-3} \text{ of water } m^{-3} \text{ of soil } s^{-1}]$ |
| 90 |                                                                                                     |
| 91 | 6) Define z in Eq. 2 and other variables in Eq. 4.                                                  |
| 92 | Response:                                                                                           |
| 93 | All terms in Equation 2 and 4 are now defined.                                                      |
| 94 |                                                                                                     |
| 95 | 7) Eqs. 17 and 18, check the third term on the RHS.                                                 |
| 96 | Response:                                                                                           |
| 97 | Third term in equation 17 and 18 is updated.                                                        |
| 98 |                                                                                                     |
| 99 | 8) Eq. 23: write cn as ci,j,k                                                                       |

**Response**:

| 101 | In equation 23, $c_n$ is now defined as $c_{n_{i,j,k}}$ . Additionally, equations 25-32 have been             |
|-----|---------------------------------------------------------------------------------------------------------------|
| 102 | updated.                                                                                                      |
| 103 |                                                                                                               |
| 104 | 9) Define ω' in Eqs. 25-31                                                                                    |
| 105 | Response:                                                                                                     |
| 106 | In equation 25-31, $\omega'$ is now replaced by $1 - \omega$ , where $\omega$ is defined as the weight in the |
| 107 | Crank-Nicholson method.                                                                                       |
| 108 |                                                                                                               |
| 109 | 10) Line 312: from Fig. 2, I see less dependence of average snow depth on topography with                     |
| 110 | SR.                                                                                                           |
| 111 | Response:                                                                                                     |
| 112 | We have fixed the typographical error and the text now reads "With SR, a much smaller                         |
| 113 | dependence of winter-average snow depth on topography is predicted"                                           |
| 114 |                                                                                                               |
| 115 | 11) How well is the 3D model developed in the paper compared to analytical solutions or                       |
| 116 | other well established numerical models?                                                                      |
| 117 | Response:                                                                                                     |
| 118 | In this work, we extended the existing 1D physics formulations for subsurface hydrologic                      |
| 119 | and thermal processes to included lateral processes. Thus, we did not compare existing                        |
| 120 | physics formulations against analytical solutions or other numerical models, but we did                       |
| 121 | ensure that lateral coupling was implemented correctly. Sanity checks were preformed to                       |
| 122 | ensure the 3D model solution is the same as in the 1D vertical model when the problem                         |
| 123 | setup is horizontally homogeneous (Results not shown).                                                        |
| 124 | The thermal model is independent of gravity. Thus, additional tests were performed                            |
| 125 | to ensure the numerical solution of the thermal model for propagation of heat is identical in                 |
| 126 | a 1D column that is oriented horizontally and vertically. A test was performed to study the                   |
| 127 | propagation of a heat perturbation that was applied on the left and top boundary of a                         |
| 128 | spatially homogeneous 2D domain (Figure 1, below). The difference of simulated                                |
| 129 | temperature between the two cases was of the order of the tolerance of the numerical                          |
| 130 | solver (Figure 1c). An additional test was performed in which a sinusodially varying                          |
|     |                                                                                                               |

131 temperature perturbation was applied on the left and top boundary; and the difference in

- results was again within tolerance of numerical solver (Figure 2). These tests ensured that
- 133 lateral coupling was correctly implemented within the model. To address the reviewer's
- 134 concerns regarding testing, we have added description of these analyses to the
- 135 Supplementary Material (Page 2, lines 18-40, and a reference to these tests has been added
- to the main text (Page 12, lines 241-244).
- 137

138

143

**139** Figure 1. Propagation of a spatially homogeneous temperature perturbation applied

140 on the (a) left and (b) top boundary of a spatially homogeneous 2D transect at the

141 end of 1-day. (c) The difference in evolved temperature between two cases is many

---

## Author Response (AR2)

November 2017

Dear Dr. Peylin,

My co-authors and I are pleased to submit our revised manuscript titled "*Impacts of microtopographic snow-redistribution and lateral subsurface processes on hydrologic and thermal states in an Arctic polygonal ground ecosystem*" for your consideration for publication in Geoscientific Model Development.

We thank you for your constructive feedbacks, which helped us to clarify important aspects of our work. Modifications made in the revised version of the manuscript are summarized below:

1. We have added text in Section 2.6 (lines 309 – 311) to acknowledge that the location of sensors does not align with 2D transect used for modeling.
2. Our work shows that explicitly resolving snow redistribution due to microtopography and lateral subsurface processes does not have an impact on domain average hydrologic and thermal states. Thus, we believe additional studies are needed to justify the need to explicitly account for fine-scale procces. Possible future extensions of our work include representing intermediate scale topographical variation and biogeochemical  cycles along with snow redistribution and lateral subsurface processes in the model. We have expanded Section 3.5 (lines 533 – 537) to recommend this type of extension of our work.
3. We have updated Figure 1 to clarify the extent of the study site.
4. Captions of Figure 4 and 6 have been updated.
5. Two missing references on Page 2 Line 56 of second submission have been added.
6. The sentence in Section 3.5 has been corrected.
7. Additionally, the U.S. Department of Energy has renamed the Accelrated Climate Modeling for Energy (ACME) to the Energy Exascale Earth System Model (E3SM). Thus, all reference to ACME and ACME Land Model (ALM) have been changed to E3SM and E3SM Land Model (ELM) in the revised manuscript.

My co-authors and I believe we have thoroughly addressed your comments and that the revised manuscript is well suited for publication in Geoscientific Model Development. We look forward to receiving your response.

Sincerely,
Gautam Bisht

[revised manuscript text omitted]